# Interpretable crop pest and disease identification based on comparative concept tree

**Bingjing Jia****[1], Zhiwei Zheng[1], Jinyu Zeng[1], Lei Shi[2], Hua Ge[1], Chenguang Song[1]***

**1** Anhui Science and Technology University, Bengbu, China, **2** State key Laboratory of Media Convergence and Communication; Communication University of China, Beijing, China

* songcg@ahstu.edu.cn

## Abstract

Deep learning provides new methods for crop pest and disease identification and control, offering unique advantages in terms of recognition accuracy and efficiency. However, deep learning models generally lack interpretability, and their internal decision-making processes are difficult to understand. This, to some extent, undermines users' trust in the model's predictions and hinders its large-scale application in agricultural production. Therefore, improving model transparency and interpretability has become an important research direction. To address this issue, this study proposes a novel interpretable crop pest and disease identification model, the Contrastive Prototype Tree (CPTR). The model is designed around the core structure of "concept prototypes and decision tree," which builds clear prototype matching paths for each recognition result. This enables the model to not only have strong classification capability but also provide intuitive explanations. Additionally, the study introduces the SimCLR contrastive learning framework to enhance the model's ability to express deep image features. SimCLR guides the model to learn more discriminative visual features by maximizing the similarity between positive sample pairs and minimizing the similarity between negative sample pairs, thereby improving overall recognition performance. This study evaluated the model on three datasets: Apple-Leaf9, Cassava, and Cashew. The experimental results show that CPTR achieves accuracies of 83.74%, 94.80%, and 96.01% on the three datasets, representing improvements of 4.12%, 0.34%, and 0.51% compared to Prototype Tree, respectively. These results indicate that the proposed model achieves the highest accuracy across different datasets, demonstrating its effectiveness.

## 1 Introduction

Globally, crop pests and diseases pose a significant challenge to agricultural production and food security [1,2]. With the intensification of global climate change and changes in agricultural production methods, the types of crop pests and diseases continue to increase, and their spread has become more complex and rapid [3,4],

**Data availability statement:** AppleLeaf9: https://github.com/JasonYangCode/AppleLeaf9. CCMT: https://www.kaggle.com/datasets/rahimanshu/ccmt-plant-disease-dataset. CPTR: https://github.com/Zhiwei563/CPTR.

**Funding:** The Anhui Science and Technology University Science Foundation (Grant No. WDRC202103, XWYJ202301), the Open Project Program of Guangxi Key Laboratory of Digital Infrastructure (No. GXDIOP2024010), the Key Project of Natural Science Research of Universities in Anhui (Grant No. 2022AH051642), the Research and Development Fund Project of Anhui Science and Technology University (Grant No. FZ230122). The funders had no role in study design, data collection and analysis, decision to publish, or preparation of the manuscript.

**Competing interests:** The authors have declared that no competing interests exist.

causing major difficulties for agricultural workers. Effective identification and early prevention of crop pests and diseases are crucial for improving crop yield and quality, reducing excessive pesticide use, minimizing environmental pollution, and ensuring food security. Efficient and accurate pest and disease identification methods can not only reduce pesticide usage but also prevent crop yield loss caused by pesticide misuse, thereby increasing crop yield while protecting the ecological environment [5,6].

Traditional identification methods mainly rely on agricultural experts' on-site observations and subjective judgment, which are not only inefficient but also significantly reduce both efficiency and accuracy when faced with large-scale production environments and diverse disease types [7]. In recent years, with the development of image recognition and computer vision, deep learning has gradually become an important technological approach for crop pest and disease identification [8,9]. Convolutional Neural Networks (CNNs), with their multi-layered nonlinear structure, can learn discriminative deep features from raw images, thereby significantly improving the accuracy of pest and disease image recognition [10].

Mainstream deep learning models often exist as "black boxes," with their internal reasoning mechanisms lacking transparency [11], making it difficult for agricultural practitioners to trust the model's output in practical applications. Especially when the model makes incorrect predictions, users often struggle to trace the decision process, analyze the error causes, and effectively locate model flaws, which reduces the model's practical value in production processes [12]. As a result, "interpretability" has gradually become a key research direction for AI models. Researchers have proposed various interpretable deep model structures, including Class Activation Maps (Grad-CAM) [13], Concept Bottleneck Networks [14], Prototype Networks [15], etc., all of which have improved the interpretability of models to some extent.

Prototype learning structures have gained widespread attention due to their strong semantic nature and clear reasoning processes. The core idea is to introduce "concept prototypes" as an intermediate bridge in classification, enabling the model not only to provide prediction results but also to highlight the semantic representative regions or samples upon which the predictions are based. Building upon this, the Concept Prototype Tree (Prototype Tree) [16] further extends the expressive capacity of prototype learning by organizing multiple concept prototypes in a hierarchical manner through a tree structure. This allows the model to make decisions along a clear logical path, progressively approaching the final classification result. This modular, staged reasoning mechanism not only enhances the model's interpretability but also improves its ability to represent complex class relationships, making it easier for users to intuitively understand the model's decision-making basis and process.

However, prototype learning still faces challenges such as blurry decision boundaries and insufficient feature representation when dealing with highly similar or widely varying image categories. SimCLR contrastive learning [17] addresses this by pulling positive sample pairs closer and pushing negative sample pairs farther apart in the feature space, enabling the model to learn more discriminative feature embeddings. This method can automatically extract structurally stable and semantically rich features from images.

Based on the above background, this paper proposes an interpretable pest and disease identification model that integrates SimCLR contrastive learning and the Concept Prototype Tree structure—Contrastive Prototype Tree (CPTR). The CPTR model is designed to balance both accuracy and interpretability. It uses CNN as the base feature extraction module and incorporates the SimCLR mechanism to enhance feature expression capabilities. Additionally, it combines a trainable prototype module with a binary concept tree structure in the decision-making process, thus achieving semantic transparency and structural explicitness in the classification path. Unlike traditional CNNs or prototype networks, CPTR not only outputs prediction results during inference but also showcases the relationships between the input image and multiple concept prototypes. This structure provides good local interpretability while offering a global decision-making context through the tree-based organization, significantly improving users' understanding and trust in the model's prediction rationale.

The structure of this paper is organized as follows: Chapter 2 presents the relevant domestic and international research progress, with a focus on reviewing the development of deep learning-based crop pest and disease identification methods and interpretable models. Chapter 3 provides a detailed description of the overall framework and key techniques of the proposed Contrastive Prototype Tree model, including dataset construction, preprocessing processes, the SimCLR contrastive learning mechanism, and the concept prototype tree structure. Chapter 4 outlines the experimental design, parameter settings, and result analysis, emphasizing the performance of CPTR across multiple crop pest and disease datasets and its interpretability validation. Chapter 5 concludes the paper, discussing the innovations, limitations, and future research directions.

The main contributions of this paper are as follows:

- This study proposes an interpretable crop pest and disease identification model based on SimCLR contrastive learning and the Concept Prototype Tree.

- To enhance the model's understanding and recognition capability of image information, the SimCLR contrastive learning mechanism is introduced to optimize feature learning, effectively improving the discriminative power and generalizability of feature expression.

- Experimental research is conducted on multiple real-world crop pest and disease image datasets. Compared with existing mainstream models, CPTR demonstrates superior overall performance.

## 2  Related work

### 2.1  Crop disease and pest identification based on deep learning

Crop pest and disease identification is a critical component in ensuring agricultural production efficiency and crop yield [18]. Traditional identification methods often rely on expert knowledge and manual experience, which not only are inefficient but also prone to subjective influences. With the development of computer vision and deep learning technologies, image-based pest and disease identification methods have gradually become mainstream.

In early studies, researchers primarily used classic convolutional neural network (CNN) architectures to classify crop pest and disease images. Fuentes et al. [19] combined three detectors—Faster R-CNN, SSD, and R-FCN—with backbone networks such as VGG-16, ResNet-50, and ResNeXt-50 to achieve real-time identification of tomato pests and diseases, maintaining high accuracy and low false positive rates even in complex environments. These object detection models are capable of locating and classifying disease areas in images, laying the foundation for practical applications in real-world scenarios.

To improve recognition ability in complex field scenarios, subsequent studies introduced drone remote sensing images and image segmentation techniques. Tetila et al. [20] used a drone-based aerial image acquisition method combined with SLIC superpixel segmentation and ResNet extractors to achieve automatic identification of soybean pest and disease

areas, with the highest classification accuracy reaching 93.82%. In addition, image augmentation and preprocessing technologies have become key methods for enhancing model performance. For example, the image processing system developed by Devaraj [21] on the MATLAB platform significantly improved overall recognition accuracy in stages such as preprocessing, feature extraction, and classification. Martos et al. [22] integrated remote sensing technology, artificial intelligence, and advanced sensor technologies to achieve efficient management and sustainable development of agricultural production.

At the same time, transfer learning has been widely applied in pest and disease identification tasks as an effective method to alleviate the scarcity of agricultural image samples. Barbedo [23] and others found that deep models, when provided with more diverse samples and high data variety, significantly improve generalization ability, highlighting the importance of building high-quality pest and disease image databases. In practical applications, mainstream models are often pre-trained on large-scale datasets like ImageNet and then fine-tuned for crop pest and disease identification tasks to enhance the model's performance on target tasks.

In summary, deep learning methods have shown promising application prospects in crop pest and disease identification. However, several issues remain in their practical application: first, obtaining high-quality annotated data in the agricultural field is difficult, which limits the training and deployment of deep models; second, existing models generally lack transparent decision logic, making it challenging to meet the need for interpretability in agricultural practice; third, the models have insufficient generalization ability across different regions or environmental conditions, leading to performance degradation.

## 2.2 Interpretable deep learning

Deep neural networks face challenges in gaining widespread trust and acceptance in agricultural applications due to the lack of transparency in their decision-making processes. Explainable Artificial Intelligence (XAI) technologies improve users' trust in model decisions by providing visual and semantic explanations of model behavior [24].

Early studies often employed post-hoc explainability methods, such as Grad-CAM, which generates activation heatmaps through gradient backpropagation, and LIME [25], which models feature contribution values through input perturbations. Shrikumar et al.'s DeepLIFT algorithm [26] can trace the impact path of input changes on the output, enhancing local interpretability. Gopalan et al. [27] proposed a maize leaf disease classification model based on ResNet152 and combined it with the Grad-CAM method to improve model interpretability. This model achieved accuracies of 99.95% in training and 98.34% in testing, effectively distinguishing between four types of maize leaf diseases.

However, post-hoc methods often lack stability and are disconnected from the model's original structure. To overcome these shortcomings, researchers have proposed structurally interpretable models, with the "Concept Bottleneck Model (CBM)" and "Concept Prototype Network (ProtoPNet)" being the most representative. CBM achieves semantic-level interpretability by constructing an intermediate layer with explicit semantic representations, where model decisions are built upon high-level concepts. ProtoPNet, on the other hand, introduces a set of class prototype images, making the classification process resemble a "this looks like that" analogy, and has achieved good interpretability results in multiple fine-grained recognition tasks. Zeng et al. [28] proposed the CDPNet model, a deformable ProtoPNet model for interpretable maize leaf disease identification. This model, by combining deformable convolution with ProtoPNet's concept prototypes, can capture more flexible and precise disease areas, thereby improving both disease diagnosis accuracy and interpretability.

The advantage of concept prototype methods lies in their intuitive reasoning logic and strong semantic associations. By using prototypes as intermediaries, these methods link input images with classes, significantly enhancing model transparency and human interpretability. For example, ProtoPNet can show, "This image belongs to apple rust disease because it closely resembles the red rust spots in this prototype image." This image-to-prototype visual mapping provides a strong local explanation basis.

However, existing concept prototype models also exhibit notable limitations. On one hand, their prototype matching mechanisms often rely on global or fixed local feature representations, making them less effective when dealing with highly similar classes or images with complex internal structures. This can lead to misclassification or ambiguous reasoning. On the other hand, most current methods do not consider the structural relationships among prototypes and therefore lack the ability to express a global decision path. As a result, although the models are interpretable, their explanations remain fragmented at the "image-to-image" level and fail to provide a complete semantic reasoning chain.

To address these issues, this study adopts the design principles of the Prototype Tree model and introduces a tree-structured representation on top of prototype learning, proposing a more hierarchical and holistic structurally interpretable framework. In this framework, multiple prototypes are organized as decision tree nodes according to semantic or discriminative pathways. The input image is matched through the tree from top to bottom, enabling multi-level reasoning that progresses from coarse to fine and from abstract to specific. Each branching decision corresponds to a prototype match and can explain why a particular path is chosen over another. This approach effectively integrates local image-level explanations with the global decision-making logic. To better compare representative interpretable deep learning methods, Table 1 summarizes the core ideas, advantages, and limitations of several widely used models.

## 3 Model

### 3.1 CPTR

The CPTR model integrates the SimCLR contrastive learning mechanism with the Concept Prototype Tree structure, balancing feature expression capability and model interpretability. The model consists of three main components: 1. CNN feature extraction layer, which is responsible for converting the input image into high-dimensional feature representations; 2. Concept Prototype Tree layer, which contains several trainable prototype nodes organized into a binary decision tree for hierarchical feature discrimination; 3. SimCLR contrastive learning and training strategy, which combines contrastive loss with classification loss as the optimization objective.

The input crop pest and disease image is passed through a convolutional neural network (CNN) to obtain the feature map $z_i$. During the training phase, this study applies random transformations to each input image to generate another view, and the same CNN is used to extract features to obtain $z_j$. SimCLR contrastive loss is then applied to push the features closer in the feature space while distancing them from features of other samples. This contrastive learning process enhances the discriminability and robustness of the features extracted by the CNN. Next, the feature map $z_i$ is passed

**Table 1. Comparison of interpretable models.**

| Model Name | Core Idea | Advantages | Disadvantages |
|---|---|---|---|
| Grad-CAM | Generates feature activation heatmaps through gradient backpropagation | Intuitive visualization, simple computation, applicable to various CNN structures | Post-hoc explanation, poor stability, cannot provide global semantic explanation |
| CBM | Introduces explicit concept layers in the model to explain the decision process using semantic variables | Provides semantic-level explanations, allows tracking of concept contributions | Concept definitions rely on manual annotations, limited generalization ability |
| ProtoPNet | Learns class prototypes for "image-prototype" analogy reasoning | Intuitive reasoning logic, provides local interpretability | Lack of structural relationships between prototypes, explanations limited to local |
| Prototype Tree | Organizes prototypes in a tree structure to establish hierarchical semantic reasoning paths | Provides global decision logic, allows hierarchical explanations | Blurry decision boundaries for highly similar categories, limited feature expression capability |
| CPTR | Combines SimCLR contrastive learning with Concept Prototype Tree structure for enhanced features and explainable reasoning | Balances classification accuracy and interpretability, strong feature discriminability, clear structural visualization | Relatively complex model structure, longer training time for contrastive learning |

through the Concept Prototype Tree layer for discrimination. The specific process is as follows: the model calculates the similarity between $z_i$ and each prototype at the tree nodes, and routes $z_i$ to the corresponding child node with a certain probability based on the similarity. In this way, the input sample propagates through the tree structure, matching the corresponding pest and disease feature prototypes layer by layer, ultimately reaching one or more leaf nodes. Due to the use of probabilistic soft routing, the input features are effectively "distributed" to each leaf node with varying probabilities. The model then performs a weighted fusion of the category predictions from each leaf node according to these probabilities to obtain the final output.

The decision tree layer in CPTR differs from traditional decision trees: it allows samples to propagate along multiple paths simultaneously in a soft branching manner, improving the model's adaptability to complex samples while maintaining differentiability. During training, the model's optimization objective is composed of both classification loss and contrastive loss. These two losses are combined using a weighted sum:

$$L_{total} = L_{cls} + \lambda L_{con} \tag{1}$$

The total loss function described above combines the supervised classification loss and the contrastive learning loss. Here, $L_{cls}$ represents the cross-entropy loss, which measures the discrepancy between the model's predictions and the ground-truth labels; $L_{con}$ denotes the SimCLR contrastive loss, which aims to enhance the discriminability of the learned feature representations; and $\lambda$ is a weighting coefficient that controls the relative contribution of the two losses. The overall architecture of the CPTR model is illustrated in Fig 1.

To examine the effect of the loss weighting coefficient $\lambda$ in Eq. (1), we conducted a small sensitivity analysis on the AppleLeaf9 dataset. Keeping all other training settings fixed, we evaluated the model performance using several values of $\lambda$ around the default setting, specifically $\lambda \in \{0.1, 0.3, 0.5, 0.7, 0.9\}$. The results show that the overall classification

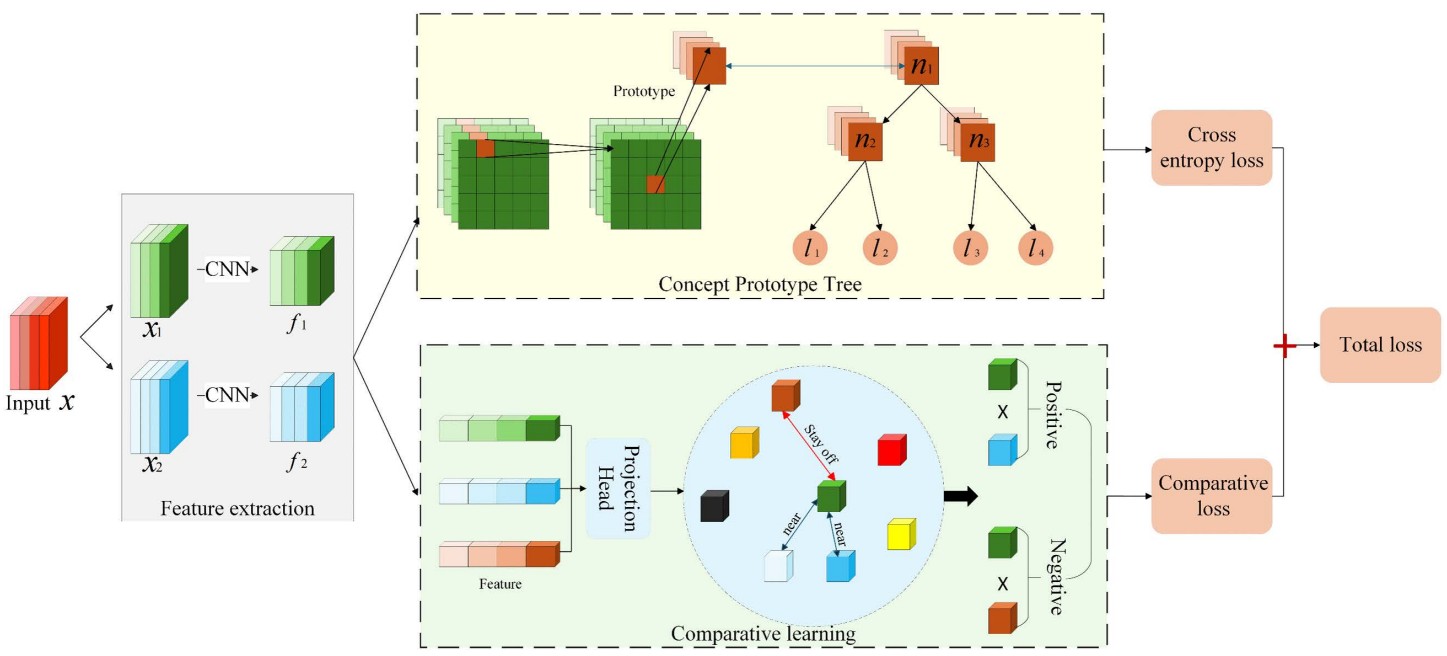

**Fig 1. CPTR model diagram.**

performance varies only slightly within this range, indicating that the model is relatively insensitive to $\lambda$ in a reasonable interval. Among these settings, $\lambda = 0.5$ provides a favorable balance between classification accuracy and feature discriminability. Therefore, $\lambda = 0.5$ is adopted as the final setting and used in all experiments reported in this paper.

## 3.2 CNN feature extraction layer

In the CPTR model, the CNN feature extraction module serves as the foundational component for image encoding, responsible for transforming the input crop pest and disease images into high-dimensional, structured visual feature representations. This module employs predefined deep convolutional neural network architectures (VGG19, ResNet152, DenseNet161) as the backbone networks, leveraging their excellent feature extraction capabilities in the field of image recognition. Let the input image be $x \in R^{C \times H \times W}$, where C is the number of channels, and $H \times W$ are the height and width of the original image. After passing through the convolutional layers and nonlinear transformations, the output feature map is denoted as:

$$z = f_\theta(x), \quad z \in R^{H \times W \times D} \tag{2}$$

Where $f_\theta(\cdot)$ represents the CNN feature extraction function, with the parameter set $\theta$, and $z$ denotes the output feature map.

Each feature map $Z_{(u,v)}$, corresponds to a semantic representation of position $(u, v)$ in the original image and contains deep structural information about the response of the convolutional kernel to that region. The entire feature map $z$ is essentially a dense representation of the input image in a high-dimensional semantic space, and is the basis for the subsequent conceptual prototype tree module for region matching.

## 3.3 Conceptual prototype tree layer

In order to enhance the interpretability of the model while ensuring the classification performance, this study introduces a conceptual prototype tree structure based on the fusion of prototype learning and soft decision tree in the CPTR model. The structure takes the conceptual prototype as the core, together with the probabilistic routing mechanism, to gradually realize the transparent inference process from feature characterization to category prediction.

In the conceptual prototype tree, each internal node n is associated with a trainable prototype $p_n$, Take $h = w = 1$, i.e., each prototype is a small $1 * 1$ patch with the number of channels D, which is the same as the number of channels in the feature map $z$. This design is able to capture fine-grained local discriminative features in the input image, such as leaf surface details, lesion distribution, etc. To determine which local region in the input feature graph $z$ is closest to the prototype $p_n$ of a node, we slide over the feature graph to extract all local patches and compute the Euclidean distance between them and $p_n$. Ultimately, the local patch with the smallest distance is selected as the optimal match:

$$\tilde{z}^* = arg \min_{\tilde{z} \in patches(z)} \| \tilde{z} - p_n \| \tag{3}$$

This optimal patch $\tilde{z}^*$ represents the local region in the input image that is most similar to the prototype, and is the basis for node decisions. Through this local matching mechanism, the model is able to perceive the key features of the input image at a fine-grained level and guide the subsequent routing direction of the samples accordingly.

After the prototype matching is completed, the routing direction of the sample in the tree needs to be determined based on the degree of matching. Specifically, the probability of the sample propagating from the current node to the right child node is defined as:

$$p_{e(n,n.right)}(z) = exp(- \| \tilde{z}^* - p_n \|) \tag{4}$$

and the probability of propagation to the left child node is its complement, viz:

$$p_{e(n,n.left)}(z) = 1 - p_{e(n,n.right)}(z) \tag{5}$$

This Soft Split strategy allows the samples not to be forcibly assigned to a single path, but to be propagated simultaneously to the left and right child nodes with a certain probability, preserving the uncertainty and informativeness of the decision-making process. In addition, the use of exponential function can naturally map the Euclidean distance to probability, ensuring that the probability of propagation to the right branch is higher when the distance is closer and the similarity is higher.

This design is different from the single routing approach in traditional hard decision trees, enabling the CPTR model to effectively handle situations with fuzzy category boundaries or large sample heterogeneity while maintaining microscopicity and facilitating end-to-end training.

After the samples are routed through multiple layers of nodes, they eventually reach each leaf node with a certain probability distribution. Let $P_\ell$ be a path from the root node to the leaf node $\ell$. The total probability of a sample reaching $\ell$ along this path is:

$$\pi_\ell(z) = \prod_{e \in P_\ell} p_e(z) \tag{6}$$

where $p_e(z)$ denotes the transfer probability of the sample corresponding to each edge on the path. The path probability $\pi_\ell(z)$ reflects the global likelihood of the sample's reasoning process in the conceptual prototype tree structure.

Each leaf node $\ell$ learns a category distribution parameter vector $c_\ell$, which is normalized by Softmax to obtain a standardized category probability distribution $\sigma(c_\ell)$. Ultimately, the category prediction of the input image is obtained by weighted summation of the outputs of all leaf nodes by path probability:

$$\hat{y}(x) = \sum_{\ell \in \mathcal{L}} \sigma(c_\ell) \cdot \pi_\ell(f(x; \omega)) \tag{7}$$

This weighted fusion mechanism not only realizes end-to-end micro-trainable model predictions but also ensures transparent traceability of the inference path and decision basis. Unlike traditional black-box neural networks, CPTR is able to provide a clear and verifiable decision-making link for each classification prediction, thus significantly improving the trust and usability of the model in real-world application scenarios such as agricultural production. The conceptual prototype tree model diagram is shown in Fig 2.

### 3.4 SimCLR contrastive learning

In order to improve the feature representation of the CPTR model, this study introduces the SimCLR contrastive learning framework for feature optimization. SimCLR obtains a more discriminative image feature representation by constructing pairs of positive and negative samples to maximize the similarity between positive samples and minimize the similarity between negative samples in the feature space.

Specifically, for a given input image x, this study first generates two different augmented views, denoted as $x_i$ and $x_j$, by means of stochastic data augmentation strategies (e.g., random cropping, random flipping, color perturbation, etc.). Both views are derived from the same original image and are therefore considered as positive sample pairs. Subsequently, the enhanced image views are fed into a convolutional neural network feature extractor with shared weights as well as a nonlinear projection head to obtain low-dimensional feature representations $z_i$ and $z_j$, respectively.

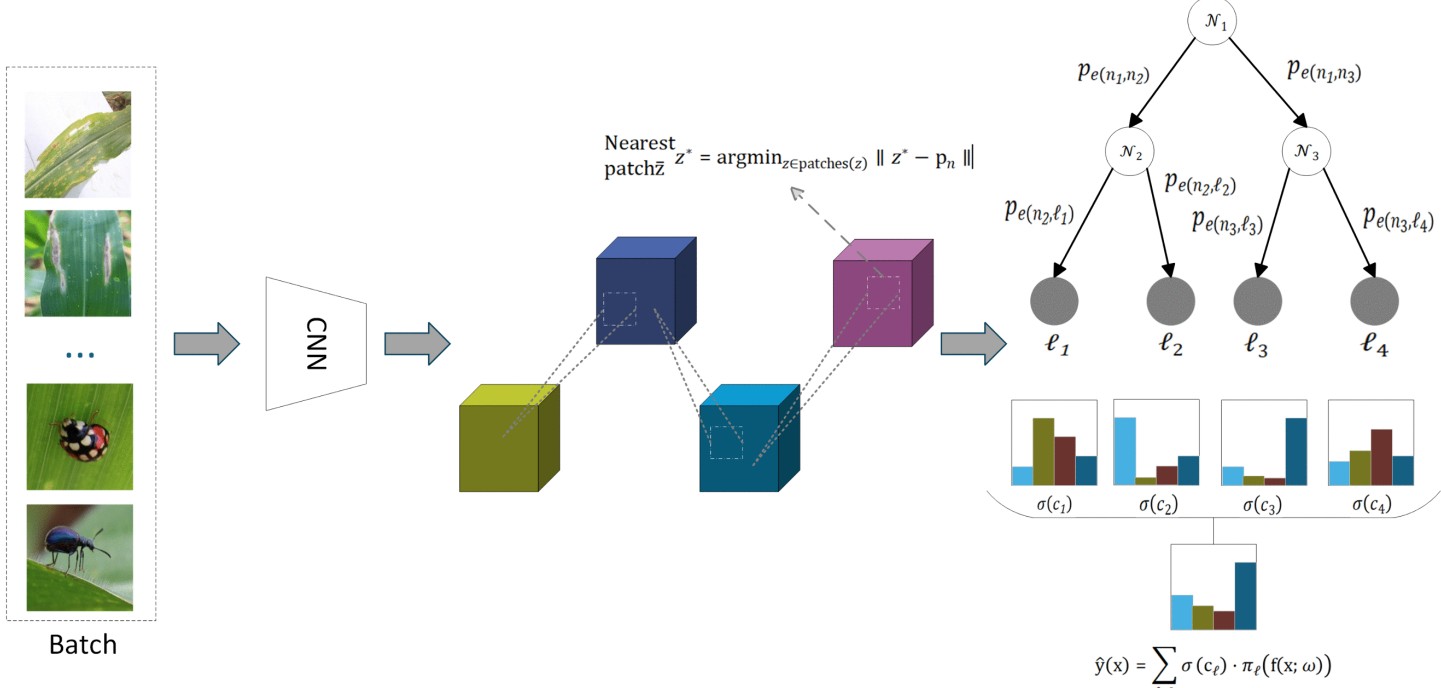

**Fig 2. Conceptual prototype tree model diagram.**

Given a batch of training samples, the SimCLR contrast loss function, is used to optimize the model parameters so that enhanced views from the same original image are closer together in feature space and features from different images are farther apart. The specific mathematical expression for the NT-Xent loss is:

$$L_{con}(z_i, z_j) = -log\frac{exp(sim(z_i, z_j)/\tau)}{\sum_{k=1}^{2N} 1_{[k \neq i]} exp(sim(z_i, z_k)/\tau)} \tag{8}$$

where $\tau$ denotes the temperature parameter and N is the number of original samples within a single training batch, $1_{[k \neq i]}$ denotes the indicator function, which is used to exclude samples of itself and to avoid comparing samples with itself in the loss calculation, $L_{con}(z_i, z_j)$ denotes the loss value of the feature representation for the comparison of $z_i$, $z_j$, $sim(z_i, z_j)$ denotes the cosine similarity between feature vectors $z_i$ and $z_j$. The contrastive learning framework is shown in Fig 3:

## 4 Experiments

### 4.1 Dataset

To systematically evaluate the recognition capability of the proposed model under different crop types and pest disease symptoms, this paper selected three publicly available and representative crop pest and disease image datasets: Cassava, Cashew, and AppleLeaf9. These datasets all originate from real field environments, featuring high resolution, clear category differences, and significant intra-class variation, which effectively reflect the complexity of actual agricultural scenarios.

To ensure fairness and reproducibility of the experimental results, all datasets were split into training and testing sets according to a unified principle. During the split process, independent random sampling was performed within each

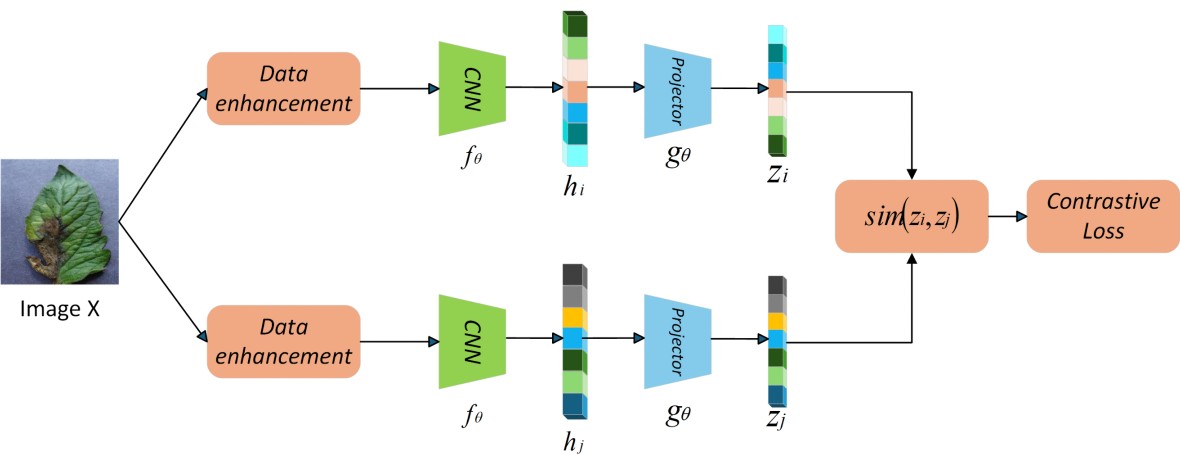

**Fig 3. Contrastive learning process diagram.**

category to ensure balanced class distribution; the random process used a fixed seed of seed = 42 to ensure consistent splitting results across different experiments. The split ratio was 80% for the training set and 20% for the testing set. The training set was used for model training and parameter updates, while the testing set was only used in the final performance evaluation phase and was not involved in any model tuning or hyperparameter selection.

The Cassava dataset is sourced from the CCMT platform [29], and primarily consists of leaf images of cassava crops collected from real field environments. It includes five categories: Healthy Leaves, Bacterial Blight, Brown Spot, Green Mite, and Mosaic. The dataset contains a total of 7,508 images, with the training and testing sets split according to a ratio while maintaining balance in class distribution. The images were collected under various lighting conditions, angles, and leaf statuses, effectively simulating the diversity and complexity of cassava field diseases.

The Cashew dataset also comes from the CCMT project and focuses on the identification of typical pests and diseases on cashew leaves. The dataset includes four categories: Healthy, Anthracnose, Red Rust, and Leaf Miner, with a total of 6,109 images. All images were manually collected and annotated by experts, with clear disease labels and high image quality, making the dataset suitable for constructing multi-class classification tasks.

The AppleLeaf9 dataset [30] is a composite dataset built for apple leaf disease identification, integrating multiple publicly available subsets, including the PlantVillage [31] database, ATLDSD, PPCD2020, and PPCD2021 [32–34]. It contains a total of 14,582 images, covering nine categories of apple leaf diseases and Healthy leaves. These categories include: Alternaria Leaf Spot, Brown Spot, Frogeye Leaf Spot, Grey Spot, Mosaic, Powdery Mildew, Rust, Scab, and Healthy Leaves. Approximately 94% of the images were captured under natural field conditions, incorporating complex factors such as uneven lighting and background interference, which greatly enhances the dataset's practical adaptability.

It should be noted that the AppleLeaf9 dataset is constructed by integrating multiple publicly available sub-datasets, and the original data do not provide unique identifiers at the leaf or scene level. As a result, when performing stratified random splitting, it is not possible to strictly guarantee that images originating from the same leaf or the same acquisition scene do not appear simultaneously in both the training and test sets. This potential sample correlation may, to some extent, lead to a slight overestimation of the overall performance.

Nevertheless, to ensure a fair comparison among different methods, all models in this study are evaluated using exactly the same data split. Under identical data conditions, the relative performance differences between models remain comparable and informative.

Detailed information about the dataset is provided in Table 2, and sample images from the dataset are shown in Fig 4.

**Table 2. Details of Cashew, Cassava and Appleleaf9 datasets.**

| Dataset | Category name | Train set | Test set | Total |
|---------|---------------|-----------|----------|-------|
| Cashew | Anthracnose | 1357 | 344 | 1701 |
| | Healthy | 1123 | 245 | 1368 |
| | Leaf Miner | 1083 | 275 | 1358 |
| | Red Rust | 1342 | 340 | 1682 |
| Cassava | Bacterial Blight | 2119 | 522 | 2614 |
| | Healthy | 954 | 238 | 1193 |
| | Brown Spot | 1185 | 296 | 1481 |
| | Green Mite | 812 | 203 | 1015 |
| | Mosaic | 964 | 241 | 1205 |
| AppleLeaf9 | Alternaria leaf spot | 318 | 99 | 417 |
| | Brown spot | 312 | 99 | 411 |
| | Frogeye leaf spot | 2569 | 612 | 3181 |
| | Grey spot | 258 | 81 | 339 |
| | Healthy | 408 | 108 | 516 |
| | Mosaic | 299 | 72 | 371 |
| | Powdery mildew | 968 | 216 | 1184 |
| | Rust | 2222 | 531 | 2753 |
| | Scab | 4312 | 1098 | 5410 |

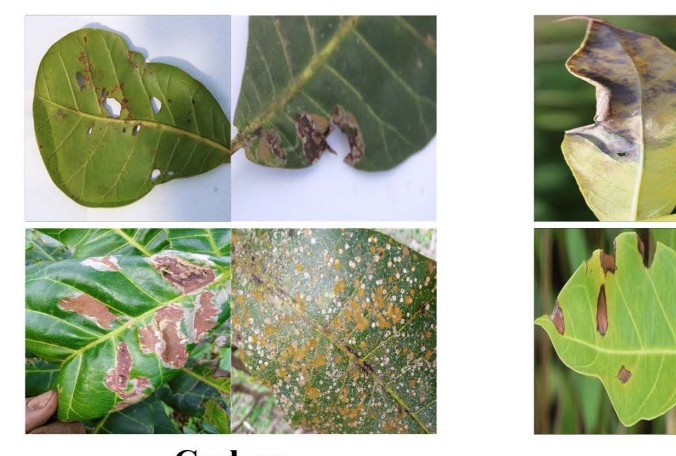
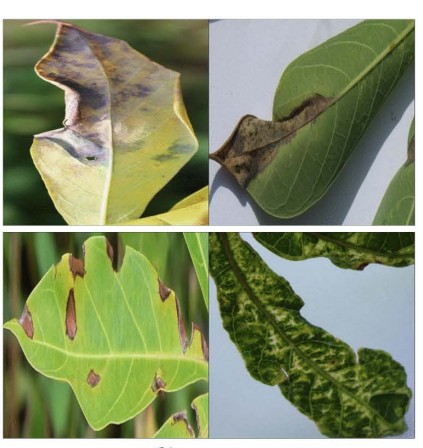
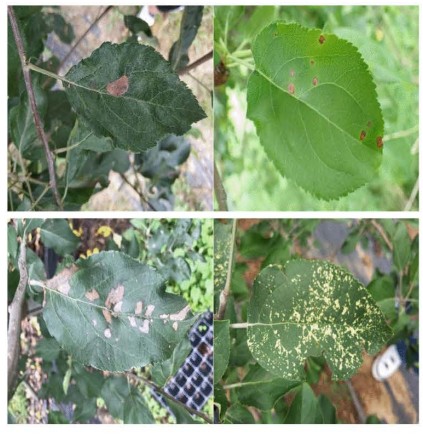

**Cashew** **Cassava** **AppleLeaf9**

**Fig 4. Partial images of Cashew, Cassava, and AppleLeaf9 datasets.**

## 4.2 Data pre-processing

To improve the model's generalization ability in crop pest and disease identification tasks and prevent overfitting during training, while also providing rich augmented sample views for the SimCLR contrastive learning module, this study applied systematic data pre-processing. First, all images were resized to 224×224 pixels to ensure consistent input dimensions. Then, two random augmented views were independently generated for each image to construct positive sample pairs for SimCLR. The specific augmentation strategies included: random perspective transformation (with a distortion_scale of 0.2, applied with a probability of 0.5) to simulate geometric deformations of leaves at different angles; color jitter (with

brightness, contrast, saturation, and hue adjustment ranges all set to 0.4, applied with a probability of 0.8) to simulate natural lighting and color variations; horizontal flipping (with a probability of 0.5) to enhance the model's robustness to orientation changes; and random affine transformations (with rotation angles of ±10°, translation scales of 0.05, shear angles of ±2°, applied with a probability of 0.8) to increase the spatial diversity of the samples.

All augmented images were normalized using the ImageNet standard for channel normalization, with mean values of [0.485, 0.456, 0.406] and standard deviations of [0.229, 0.224, 0.225] for the RGB channels, ensuring the stability of feature distribution. This data pre-processing strategy significantly improved the model's robustness under varying lighting, angles, and background conditions, and provided sufficient view diversity for SimCLR contrastive learning, thereby enhancing the discriminability and generalization capability of the feature representations. As shown in Fig 5, the SimCLR-based data augmentation strategy generates two distinct augmented views of each original image through a sequence of stochastic transformations, providing the foundation for contrastive representation learning.

### 4.3 Experimental parameters

In this study, all experiments were conducted at a resolution of 224×224. During the first 30 epochs of training, the backbone network parameters were frozen, and only the concept prototype layer and SimCLR projection head were optimized to ensure stable convergence of the feature extraction part. Afterward, in the remaining 70 epochs, all parameters were unfrozen for end-to-end joint optimization.

The training was conducted for a total of 100 epochs, with a batch size set to 64. The Adam optimizer was used with parameters $\beta_1 = 0.9$, $\beta_2 = 0.999$, and weight decay set to $1 \times 10^{-4}$. A layered learning rate strategy was employed for different modules: a smaller learning rate of $1 \times 10^{-5}$ was used for the backbone network (CNN backbone) to avoid

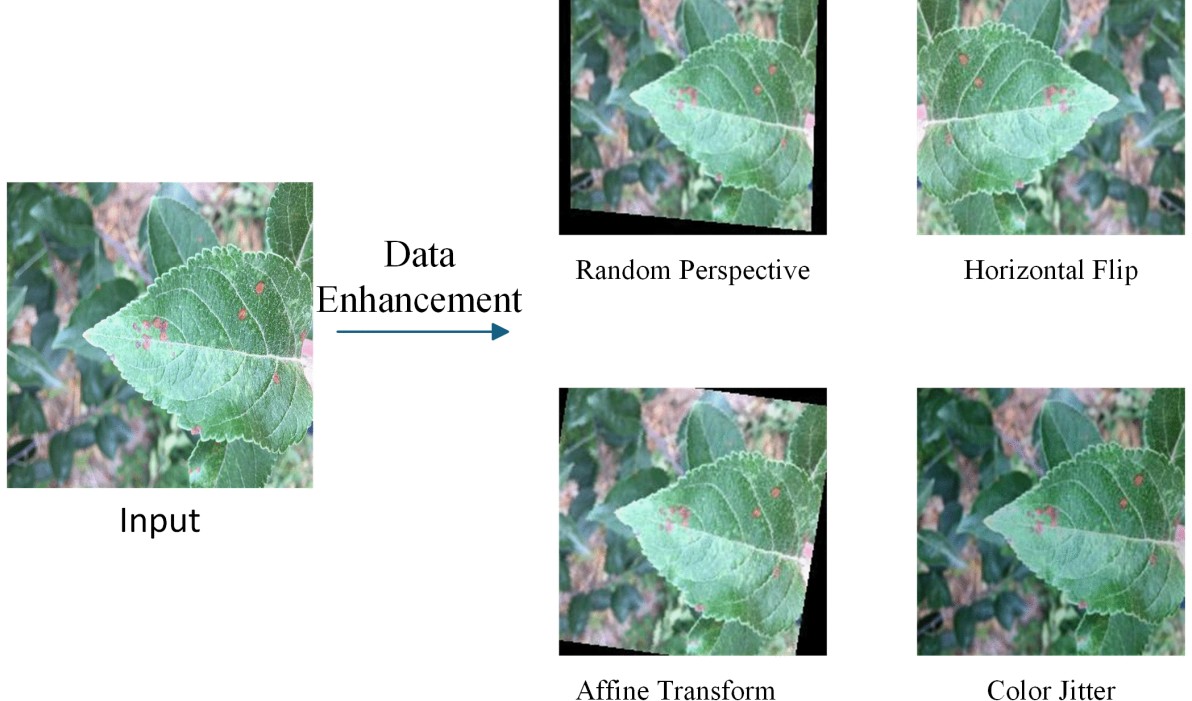

**Fig 5. Schematic diagram of data augmentation.**

disrupting the pretrained weights, while a primary learning rate of 0.001 was applied to the concept prototype layer and projection head to accelerate the convergence of the new feature space.

The learning rate scheduling adopted a milestone-based decay strategy, where the learning rate was multiplied by a decay factor of 0.1 every 10 epochs starting from epoch 60. All hidden layers used ReLU as the activation function. The concept layer nodes used the Sigmoid activation function to maintain the independence of semantic concept responses, while the final classification output layer used the Softmax function for normalization to achieve multi-class pest and disease identification. The channel dimension of the concept prototypes was set to D = 256, and the decision tree depth was set to 3.

In the SimCLR contrastive learning module, the temperature parameter $\tau$ was set to 0.5. The projection head used a two-layer multi-layer perceptron structure: the first layer was Linear(256 → 2048) followed by Batch Normalization and ReLU activation, and the second layer was Linear(2048 → 128). The output vector was L2-normalized before loss calculation to compute the cosine similarity. The similarity in the SimCLR branch was calculated using cosine distance, while the concept prototype tree module's prototype matching was based on Euclidean distance. This optimization configuration maintained the stability of the pre-trained features while ensuring the fast convergence of newly introduced modules and the overall stability of the training. It effectively balanced the model's performance in terms of classification accuracy, feature expression capability, and interpretability.

## 4.4 Evaluation indicators

To evaluate the effectiveness of the proposed model, CPTR is compared with Prototype Tree, VGG19 [35], ResNet152 [36], DenseNet161 [37], Vision Transformer [38], and Swin Transformer [39] on three datasets. All methods adopt the same data augmentation strategy, and their performance is assessed using four commonly used metrics: accuracy, precision, recall, and F1 score.

To ensure a fair and controlled comparison, all compared methods are evaluated under identical pretraining conditions. Specifically, the backbone networks of all models—including standard CNN baselines (VGG19, ResNet152, DenseNet161), Vision Transformer, Swin Transformer, Prototype Tree, and the proposed CPTR—are initialized using weights pretrained on the iNaturalist 2017 dataset.

By adopting a unified pretraining strategy across different model families, the influence of pretraining data is effectively controlled. Under this setting, performance differences among methods can be attributed to differences in model architectures and training mechanisms rather than advantages arising from pretraining.

Within the prototype-based framework, CPTR and Prototype Tree further share the same backbone architecture and identical pretraining initialization. Therefore, the performance improvement achieved by CPTR over Prototype Tree reflects the contribution of the proposed SimCLR-based contrastive learning mechanism. This interpretation is further supported by the ablation study reported in Section 4.6, where the effect of SimCLR is examined while keeping all other settings unchanged.

For clarity, all reported results (Accuracy, macro-averaged Precision, Recall, and F1) are presented as mean ± standard deviation over three independent runs with different random seeds.

Accuracy is the proportion of correctly categorized samples to the total sample.

$$Accuracy = \frac{TP + TN}{TP + TN + FP + FN} \tag{9}$$

The precision rate is the proportion of samples correctly predicted to be in the positive category to all samples predicted to be in the positive category.

$$Precision = \frac{TP}{TP + FP} \tag{10}$$

Recall is the proportion of samples correctly predicted to be in the positive category to all samples that are actually in the positive category.

$$Recall = \frac{TP}{TP + FN}$$

(11)

The F1 score, which is a reconciled average of precision and recall, is a composite metric that is particularly applicable in cases of category imbalance.

$$F1 = 2 \times \frac{Precision \times Recall}{Precision + Recall}$$

(12)

Among them, TP (True Positive) is true positive, TN (True Negative) is true negative, FP (False Positive) is false positive and FN (False Negative) is false negative.

For multi-class classification, Precision/Recall/F1 are computed in a macro-averaged manner by treating each class as one-vs-rest and averaging across classes

### 4.5 Comparative experiment

To comprehensively validate the effectiveness of the CPTR model proposed in this paper, comparative experiments were conducted on three crop pest and disease datasets: AppleLeaf9, Cassava, and Cashew. All results are the averages of three independent runs. The comparison models include traditional convolutional neural network models (VGG19, ResNet152, DenseNet161) as well as recently outstanding Vision Transformer and Swin Transformer models, to ensure the comprehensiveness and objectivity of the evaluation. The experimental results are shown in Table 3.

Overall, while traditional CNN models and transformer-based models performed well on some datasets, they still showed certain gaps compared to the CPTR model. Specifically, on the most complex AppleLeaf9 dataset, which has subtle inter-class differences, CPTR demonstrated a significant advantage: CPTR-vgg19 achieved an accuracy of 83.74%, which is nearly 6% higher than the corresponding VGG19 model, and showed comprehensive improvements in precision, recall, and F1 score. This indicates that contrastive learning effectively enhanced the model's ability to capture fine-grained disease features. On the Cassava dataset, CPTR-DenseNet161 achieved an accuracy of 94.80%, outperforming all baseline models, including the relatively strong Swin Transformer (94.73%), further confirming that CPTR retains its advantages even in conditions with clear class boundaries. On the Cashew dataset, CPTR-DenseNet161 achieved the highest accuracy of 96.01%, also leading in other metrics compared to all other models, showcasing its high recognition stability and generalization ability.

In summary, CPTR outperformed traditional CNN and transformer models across all three datasets, with a particular advantage in fine-grained recognition tasks. This success is attributed to CPTR's combination of SimCLR contrastive learning and the Concept Prototype Tree structure, which enhances the model's feature expression ability while maintaining strong interpretability and robustness.

### 4.6 Ablation experiment

To further analyze the model's performance, this paper compares the classification results of Prototype Tree and CPTR on the three crop pest and disease datasets before and after the introduction of the SimCLR contrastive learning module. The experimental results are shown in Table 4 (using DenseNet161 for feature extraction as an example). The introduction of SimCLR had a positive impact on the model's accuracy and other evaluation metrics.

On the AppleLeaf9 dataset, CPTR's accuracy increased from 78.66% to 80.48%, with significant improvements in precision, recall, and F1 score, indicating that contrastive learning enhanced the model's discriminative ability for complex

**Table 3. Comparison of Experimental Results (mean ± std over three independent runs).**

| Dataset | Model | Accuracy | Precision | Recall | F1 score |
|---|---|---|---|---|---|
| AppleLeaf9 | VGG19 | 77.43±0.74 | 77.10±0.69 | 78.19±0.81 | 77.65±0.72 |
| | ResNet152 | 76.85±0.67 | 76.50±0.73 | 76.85±0.78 | 76.65±0.70 |
| | DenseNet161 | 76.47±0.71 | 76.66±0.64 | 76.47±0.75 | 76.54±0.68 |
| | Vision Transformer | 79.12±0.58 | 77.76±0.59 | 80.91±0.55 | 77.88±0.60 |
| | Swin Transformer | 80.69±0.60 | 82.14±0.47 | 80.71±0.54 | 79.97±0.45 |
| | CPTR-VGG19 | 83.74±0.68 | 83.56±0.61 | 83.95±0.69 | 83.66±0.64 |
| | CPTR-ResNet152 | 80.00±0.69 | 80.09±0.54 | 81.99±0.73 | 80.82±0.67 |
| | CPTR-DenseNet161 | 80.48±0.63 | 80.44±0.53 | 81.30±0.71 | 80.70±0.71 |
| Cassava | VGG19 | 85.40±0.73 | 85.46±0.78 | 85.98±0.86 | 85.65±0.74 |
| | ResNet152 | 91.87±0.66 | 91.98±0.70 | 91.07±0.70 | 91.65±0.68 |
| | DenseNet161 | 81.73±0.91 | 82.39±0.87 | 81.73±0.93 | 82.23±0.89 |
| | Vision Transformer | 93.93±0.55 | 93.91±0.53 | 94.02±0.57 | 93.94±0.54 |
| | Swin Transformer | 94.73±0.45 | 95.01±0.48 | 95.02±0.44 | 95.00±0.49 |
| | CPTR-VGG19 | 92.33±0.62 | 92.86±0.58 | 92.80±0.68 | 92.82±0.66 |
| | CPTR-ResNet152 | 93.93±0.60 | 93.93±0.55 | 94.12±0.56 | 94.02±0.53 |
| | CPTR-DenseNet161 | 94.80±0.60 | 94.86±0.51 | 94.84±0.52 | 94.62±0.48 |
| Cashew | VGG19 | 91.17±0.71 | 91.54±0.84 | 91.10±0.90 | 91.44±0.86 |
| | ResNet152 | 94.67±0.65 | 94.73±0.57 | 95.02±0.62 | 95.04±0.58 |
| | DenseNet161 | 89.09±0.80 | 89.25±0.83 | 89.09±0.97 | 88.96±0.94 |
| | Vision Transformer | 95.92±0.47 | 95.80±0.40 | 96.11±0.49 | 95.94±0.46 |
| | Swin Transformer | 95.87±0.44 | 96.08±0.41 | 96.18±0.46 | 96.13±0.43 |
| | CPTR-VGG19 | 95.75±0.46 | 95.38±0.48 | 95.50±0.47 | 95.20±0.49 |
| | CPTR-ResNet152 | 95.85±0.43 | 95.77±0.45 | 96.00±0.44 | 95.64±0.46 |
| | CPTR-DenseNet161 | 96.01±0.39 | 95.82±0.41 | 95.76±0.43 | 95.65±0.42 |

disease features. On the Cassava dataset, although Prototype Tree already showed high performance, after introducing SimCLR, CPTR still achieved improvements in accuracy, precision, recall, and F1 score, further validating the effectiveness of contrastive learning in enhancing feature expression. On the Cashew dataset, CPTR also achieved a small performance improvement compared to the baseline model, indicating that SimCLR contrastive learning provides consistent optimization across different datasets.

Overall, the addition of SimCLR made the model more robust and discriminative across all evaluation metrics, providing higher accuracy and interpretability for deep learning models in crop pest and disease identification tasks.

### 4.7 Confusion matrix

This paper conducts a confusion matrix analysis of the CPTR model's prediction results on the AppleLeaf9, Cassava, and Cashew datasets. The confusion matrix provides an intuitive way to reflect the model's classification accuracy across different categories, as well as the specific category combinations that are prone to confusion. It is an essential tool for evaluating the model's fine-grained recognition ability.

From the Fig 6 results, it can be observed that CPTR exhibits a highly concentrated diagonal distribution across all three datasets, indicating that the model consistently classifies the majority of the samples correctly. For categories with a large number of samples or distinct feature patterns (such as Mosaic in Cassava, Healthy in Cashew, and the categories in AppleLeaf9 with clear lesion features), the model shows near-perfect classification performance, with very few misclassifications. In contrast, some categories with high similarity or ambiguous boundaries, such as the similar-looking Brown

**Table 4. Results of ablation experiment.**

| Dataset | SimCLR | Prototype Tree | Accuracy | Precision | Recall | F1 score |
|---|---|---|---|---|---|---|
| AppleLeaf9 | × | ✓ | 78.66 | 77.50 | 77.21 | 77.25 |
| | ✓ | ✓ | 80.48 | 80.44 | 81.30 | 80.70 |
| Cassava | × | ✓ | 94.46 | 94.65 | 94.76 | 94.56 |
| | ✓ | ✓ | 94.80 | 94.86 | 94.84 | 94.62 |
| Cashew | × | ✓ | 95.50 | 95.52 | 95.56 | 95.51 |
| | ✓ | ✓ | 96.01 | 95.82 | 95.76 | 95.65 |

Spot and Grey Spot in AppleLeaf9 (classes 8 and 9), or certain spot diseases in Cashew, still show a small amount of confusion. However, the overall error rate is noticeably lower than that of Prototype Tree and other comparison models, indicating that CPTR has stronger feature expression capability for fine-grained distinctions.

At the same time, after the inclusion of SimCLR, CPTR demonstrated clearer inter-class boundaries across multiple datasets, with misclassifications being more concentrated in a few hard-to-differentiate categories. There were no systemic biases or large-scale misclassifications. This suggests that contrastive learning effectively enhanced the model's sensitivity to key lesion textures, shapes, and color variations, improving the model's robustness in complex disease scenarios.

In conclusion, the confusion matrix analysis further confirms that CPTR exhibits stable and reliable classification ability in crop pest and disease identification tasks.

### 4.8 Interpretable analysis

In addition to its accuracy advantage, CPTR also offers excellent model decision interpretability. As shown in Fig 7, CPTR uses the Concept Prototype Tree structure to visually present the model's decision-making process in a tree-like form. The figure illustrates the concept prototype decision tree trained on the Cassava dataset (tree height h = 3), where the image at each node represents a concept prototype, and the leaf nodes correspond to specific pest and disease categories. Through this tree-based prototype display, the model's decision path is clearly revealed: during each binary decision, the model progresses along different branches based on the "presence" or "absence" of certain feature patterns in the input image, until it reaches the leaf node to give the final classification result. Each path from the root node to the leaf node corresponds to a human-understandable decision logic.

To further enhance the interpretability of the tree model, CPTR pruned the prototype tree after training: redundant branches with unclear decision boundaries were removed, leaving only prototype nodes with more certain decisions. After pruning, the model's prediction accuracy remained almost unchanged, but the number of concept prototypes in the tree was significantly reduced, and the decisions at each leaf node became clearer. By visualizing the concept tree, users can intuitively understand how the model gradually makes decisions based on specific image features, greatly increasing trust in the model's predictions.

In summary, the CPTR model not only achieves high-accuracy recognition but also provides transparent decision-making grounds by integrating the prototype tree structure, striking a good balance between performance and interpretability in crop pest and disease identification tasks.

As illustrated in Fig 8, the input image is first processed by the backbone network to extract feature representations, which are then propagated through the prototype tree along a similarity-driven path. At each internal node, the model selects the most relevant branch based on the matching relationship between the current feature representation and the corresponding prototype, thereby progressively narrowing the decision space. This process intuitively reflects the model's hierarchical discrimination from low-level local visual patterns to high-level semantic concepts. Finally, the sample reaches

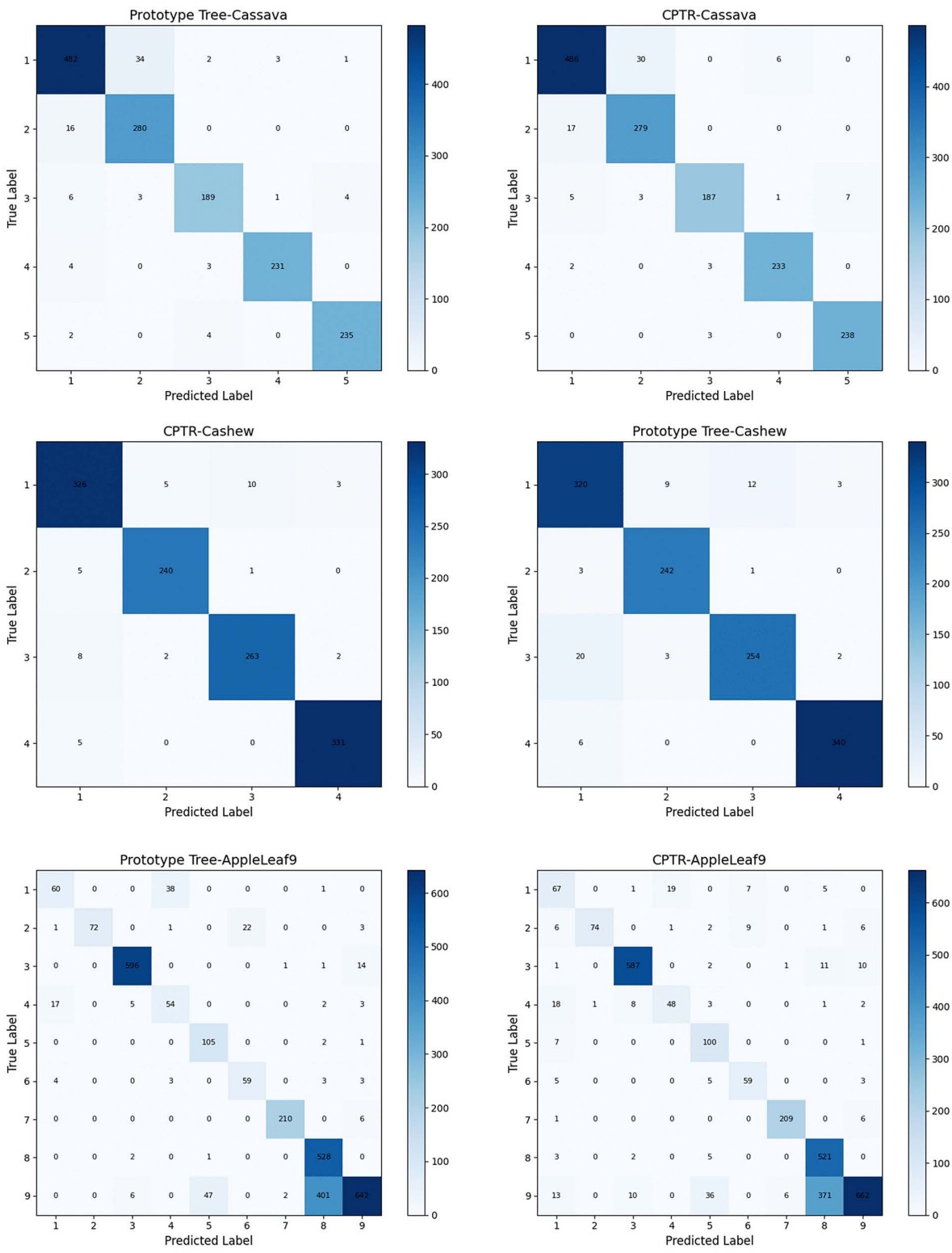

**Fig 6. Confusion matrix.**

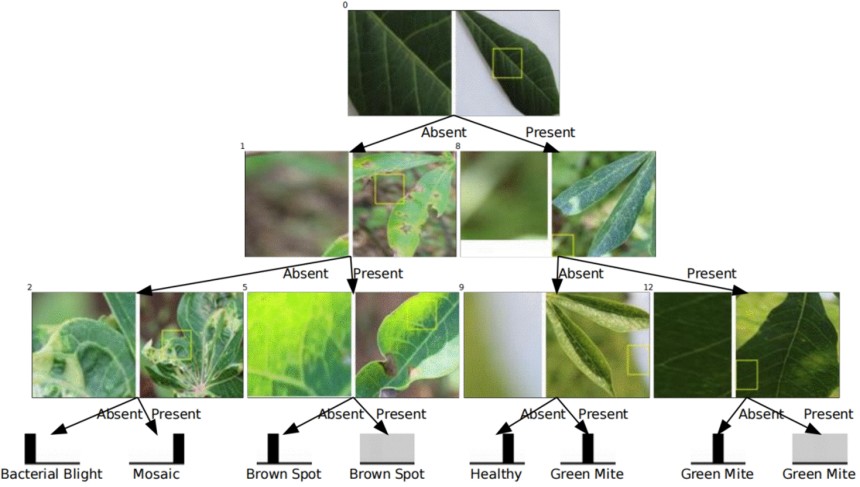

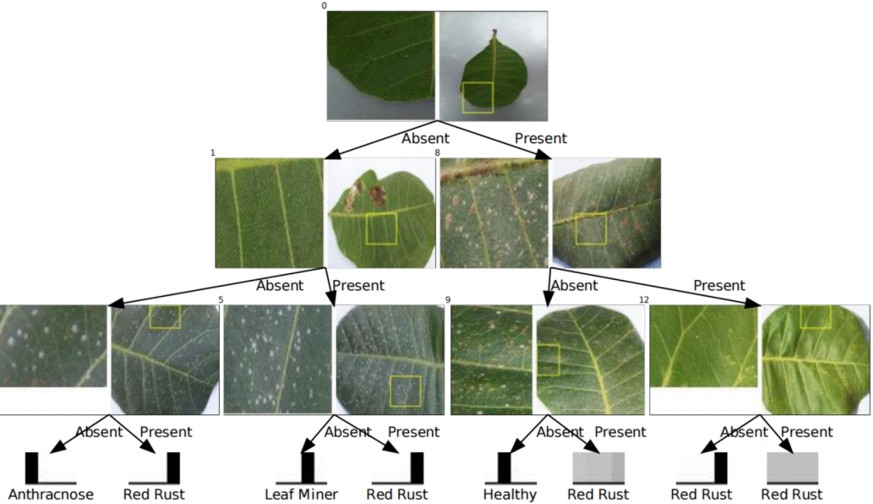

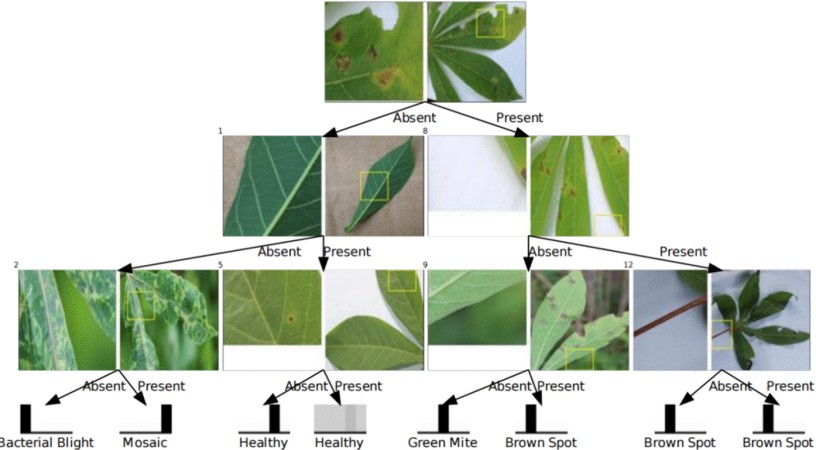

**Fig 7. Decision tree trained by CPTR.**

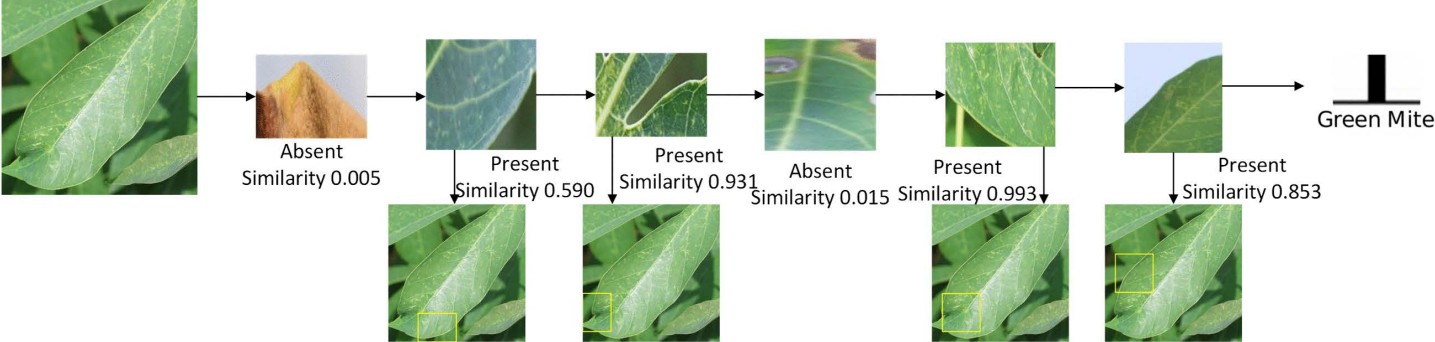

**Fig 8. An illustrative decision path of the CPTR model for a single sample.**

a leaf node and outputs the predicted class, which is consistent with the class associated with the sequence of prototype matches along the decision path. This example demonstrates how the prototype-based reasoning process directly supports the model's final decision, providing an interpretable explanation for an individual prediction while maintaining strong classification performance.

To quantitatively evaluate the consistency between the model's prediction and its interpretable decision process, we adopt the Fidelity metric. For an input sample $x_i$, $y_{model}(x_i)$ denote the final predicted class produced by the CPTR model, and let $y_{tree}(x_i)$ denote the class inferred from the corresponding prototype tree decision path, i.e., the class associated with the leaf node selected by the prototype tree for sample $x_i$. Fidelity is defined as

$$Fidelity = \frac{1}{N}\sum_{i=1}^{N}\mathbb{I}(y_{model}(x_i) = y_{tree}(x_i))$$

(13)

where N is the number of test samples and $\mathbb{I}(\cdot)$ is the indicator function, which equals 1 if its argument is true and 0 otherwise. A higher Fidelity value indicates stronger agreement between the model's final output and the reasoning outcome provided by the prototype tree, reflecting better interpretability consistency. In other words, Fidelity measures how often the final prediction of CPTR matches the class implied by its prototype-tree decision path, thereby quantifying the consistency between the model's prediction and its interpretable reasoning process.

The experimental results are shown in Table 5. The results show that CPTR achieved higher fidelity on both the Apple and Cassava datasets (97.42% and 98.80%, respectively), showing an improvement over Prototype Tree. This indicates that after incorporating contrastive learning, the features learned by the model are more robust, and the decision path is more consistent with the prototype matching process. On the Cashew dataset, CPTR's fidelity is slightly lower than that of Prototype Tree, but it still remains at a very high level, suggesting that its interpretability is still reliable.

## 5 Conclusion

In this paper, for the problem of insufficient model interpretability in crop pest and disease identification, we propose an interpretable identification algorithm that integrates SimCLR contrastive learning with the conceptual prototype tree structure —— Contrastive Prototype Tree (CPTR). The algorithm utilizes SimCLR contrastive learning to enhance feature expressiveness, and enhances the transparency of the decision-making process through the conceptual prototype tree structure, which improves the interpretability of the model while ensuring recognition accuracy. In this study, CPTR was experimentally evaluated on three crop pest and disease image datasets, AppleLeaf9, Cashew, and Cassava, and the results showed that the model was able to provide global decision interpretability through the tree structure while

**Table 5. Comparison of fidelity across different models on datasets.**

| Dataset | Model | Fidelity (%) |
|---|---|---|
| apple | Prototype Tree | 96.30 |
| | CPTR | 97.42 |
| cassava | Prototype Tree | 98.73 |
| | CPTR | 98.80 |
| cashew | Prototype Tree | 99.91 |
| | CPTR | 99.25 |

maintaining a high level of accuracy and giving a clear explanation of the decision path for each prediction. In summary, CPTR demonstrates excellent performance and good interpretability in the crop pest recognition task, providing strong technical support and new research ideas for the research of interpretable deep learning algorithms in agriculture.In future work, we will further validate CPTR on more diverse, large-scale field datasets and explore lightweight architectures to support deployment on edge devices for real-time diagnosis.

## Author contributions

**Conceptualization:** Zhiwei Zheng.

**Data curation:** Zhiwei Zheng, Hua Ge.

**Formal analysis:** Zhiwei Zheng.

**Methodology:** Bingjing Jia, Zhiwei Zheng, JinYu Zeng.

**Project administration:** Hua Ge.

**Resources:** Bingjing Jia, Hua Ge, Chenguang Song.

**Software:** Bingjing Jia, Zhiwei Zheng.

**Supervision:** Bingjing Jia, Chenguang Song.

**Validation:** Bingjing Jia, Zhiwei Zheng, JinYu Zeng, Lei Shi.

**Visualization:** Bingjing Jia, Zhiwei Zheng.

**Writing – original draft:** Bingjing Jia.

**Writing – review & editing:** Bingjing Jia.

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
