## [Decision Letter · Decision Letter 0]

6 Oct 2025

Dear Dr. Jia,

plosone@plos.org. A rebuttal letter that responds to each point raised by the academic editor and reviewer(s). You should upload this letter as a separate file labeled 'Response to Reviewers'.A marked-up copy of your manuscript that highlights changes made to the original version. You should upload this as a separate file labeled 'Revised Manuscript with Track Changes'.An unmarked version of your revised paper without tracked changes. You should upload this as a separate file labeled 'Manuscript'.

We look forward to receiving your revised manuscript.

Kind regards,

Tomo Popovic, Ph.D.

Academic Editor

PLOS ONE

Journal Requirements:

“the Anhui Science and Technology University Science Foundation (Grant No. WDRC202103, XWYJ202301), the Open Project Program of Guangxi Key Laboratory of Digital Infrastructure (No. GXDIOP2024010), the Key Project of Natural Science Research of Universities in Anhui (Grant No. 2022AH051642), the Research and Development Fund Project of Anhui Science and Technology University (Grant No. FZ230122).”

6. We note that your Data Availability Statement is currently as follows: All relevant data are within the manuscript and in Supporting Information files.

Additional Editor Comments:

PLOS ONE requires that all underlying data be made publicly available via a Data Availability Statement (DAS). Please add a dedicated DAS with persistent identifiers (e.g., Mendeley Data DOI for CCMT, GitHub link for AppleLeaf9). Ensure this appears in the manuscript text.All code and materials needed to replicate computational work should be publicly available. Deposit code, preprocessing scripts, trained weights, and environment details in a public repository (e.g., GitHub, Zenodo) and reference this in the manuscript.Manuscripts must include sufficient detail to enable replication, with transparent methods and analysis protocols.Provide a full reproducible protocol, algorithm and/or pseudocode, including validation strategy, random seeds, repeated experiments with variance measures, and all missing details.Analyses must be sound, transparent, and statistically rigorous. Report results over multiple runs with measures of variability (meansm SDs and/or CIs).Authors must confirm ethical compliance and proper attribution of third-party materials. Add a short ethics statement confirming use of publicly available datasets under their respective licenses and acknowledge licenses in figure captions if sample images are shown.Conclusions must be supported by evidence, and claims must be appropriately moderated. Either add direct, reproducible comparisons with these methods or revise claims to describe results as “competitive” rather than “state-of-the-art.”

Reviewers' comments:

Reviewer's Responses to Questions

**Comments to the Author**

1. Is the manuscript technically sound, and do the data support the conclusions?

Reviewer #1: Yes

Reviewer #2: Partly

2. Has the statistical analysis been performed appropriately and rigorously?

Reviewer #1: Yes

Reviewer #2: No

3. Have the authors made all data underlying the findings in their manuscript fully available?

Reviewer #1: Yes

Reviewer #2: No

4. Is the manuscript presented in an intelligible fashion and written in standard English?

Reviewer #1: Yes

Reviewer #2: No

Reviewer #1: The authors proposed a new model Comparative Concept Tree for interpretable crop pest and disease identification, which combines Conceptual prototype trees to provide transparent, hierarchical decision paths and SimCLR contrastive learning to enhance feature representation and discrimination. Evaluation on three datasets shows improvements over baselines and existing prototype tree approaches, both in accuracy and interpretability.

Please find below my comments

- Please quantify contributions in Abstract.

- At the end of Introduction please provide an overview of the paper by sections.

- In section 2 it would be of interest to the readers to have Tables that summarizes related work and advantages and disadvantages of proposed model in the paper.

- Please check grammar/formatting issues (missing spaces before citations, inconsistent figure captions).

- The chosen baselines are quite good, but recent transformer-based models (e.g., Vision Transformers, Swin Transformer) are not included. This limits the novelty claim. Please comment.

- Interpretability is illustrated qualitatively (decision tree visualization), but no user study or quantitative interpretability metric (e.g., faithfulness, comprehensibility scores) is provided.

- It is unclear how much each component (prototype tree vs. SimCLR) contributes to improvements. A controlled ablation experiment would strengthen claims.

- References are Comprehensive, but mostly up to 2021. Adding 2022–2024 references on interpretable deep learning in agriculture would strengthen the related work.

- Please provide future work.

Reviewer #2: Thank you for an interesting manuscript. The idea of combining prototype based reasoning with a soft decision tree and a SimCLR-style contrastive objective for plant-disease classification is well motivated. I especially appreciate the path visualizations through the tree, which help explain why the model reaches a decision. Results on AppleLeaf9, Cashew, and Cassava are encouraging and suggest practical promise.

What I still miss, to fully stand behind the conclusions, is a stronger foundation for reproducibility and fair comparison. Please provide concrete details for the contrastive component: the exact temperature (τ), the architecture and dimensionality of the projection head, the full augmentation recipe with parameter values and probabilities, and normalization choices. The joint loss weighting between classification and contrastive terms is introduced but not specified—state the value you use, how it was selected, and add a brief ablation/sensitivity analysis.

The optimization setup also needs to be unambiguous. Two learning rates are mentioned, but it’s unclear which applies to the backbone and which to prototypes/projection head; the scheduler is described qualitatively, weight decay and optimizer β/β1/β2 (or momentum) are not reported. Please provide a single, end-to-end training “recipe” that another group can follow without guesswork.

For the datasets, a transparent account of how train/val/test splits were constructed and which random seeds were used is essential. This is particularly important for AppleLeaf9, which aggregates multiple sources: without provenance-aware splits, leakage (the same scene/leaf across splits) is a real risk. Ideally, include scripts that deterministically recreate the splits.

Regarding comparisons, I would like clear evidence that baseline models were trained under exactly the same conditions (same pretraining, augmentations, scheduler, image size, and compute budget). Without that symmetry, performance gaps are hard to interpret. At present the results are single-point estimates; please add repeated runs with different seeds and report mean ± SD or confidence intervals. Given class imbalance, macro-averaged metrics and per-class precision/recall/F1 should also be reported.

The interpretability story is appealing but remains qualitative. A few lightweight quantitative indicators would substantiate the claims: prototype purity (class consistency per prototype), path entropy/consistency through the tree, and distances/similarities to retrieved prototypes along the chosen route. Two or three worked examples:input,nearest prototypes at each node, branching probabilities, final decision - would greatly help readers.

On presentation, a round of language and typesetting edits would improve clarity, there are a few typographical issues and symbols that do not render cleanly in formulas. In one place you mention a Sigmoid while the task is standard multi-class (Softmax is expected); please align terminology to avoid confusion.

The Data Availability statement needs correction. Because you rely on public datasets, please cite the exact sources/links, and—consistent with PLOS ONE policy—release code and trained weights (GitHub etc). That will address the main reproducibility concerns.

This is a good idea with practical potential. If you complete the training details, ensure fair and statistically grounded comparisons, add a small quantitative slice of the interpretability evaluation, and release code/models with an updated availability statement, the manuscript could become a strong contribution. I recommend Major Revision.

what does this mean?). If published, this will include your full peer review and any attached files.

**Do you want your identity to be public for this peer review?** For information about this choice, including consent withdrawal, please see our Privacy Policy

Reviewer #1: No

Reviewer #2: No

---

## [Author Response · Author response to Decision Letter 1]

20 Nov 2025

Summary of Changes

This document outlines the additional information provided in addressing the reviewers’ comments. The following numbering list clearly outlines how we have addressed all the comments raised by each of the two reviewers.

We wish to express sincere thanks to the reviewers for their comprehensive, detailed and insightful comments. We have taken the observations on board and we feel that this input has greatly added to the quality of the paper.

Reviewer 1:

Comment 1: Please quantify contributions in Abstract.

Response 1: Thank you for your comments. We have made improvements to the abstract section.

Action 1: We have revised the manuscript according to the opinions of the reviewer. In the revised version, we have added the following content:

Abstract:Deep learning provides new methods for crop pest and disease identification and control, offering unique advantages in terms of recognition accuracy and efficiency. However, deep learning models generally lack interpretability, and their internal decision-making processes are difficult to understand. This, to some extent, undermines users' trust in the model's predictions and hinders its large-scale application in agricultural production. Therefore, improving model transparency and interpretability has become an important research direction. To address this issue, this study proposes a novel interpretable crop pest and disease identification model, the Contrastive Prototype Tree (CPTR). The model is designed around the core structure of "concept prototypes and decision tree," which builds clear prototype matching paths for each recognition result. This enables the model to not only have strong classification capability but also provide intuitive explanations. Additionally, the study introduces the SimCLR contrastive learning framework to enhance the model's ability to express deep image features. SimCLR guides the model to learn more discriminative visual features by maximizing the similarity between positive sample pairs and minimizing the similarity between negative sample pairs, thereby improving overall recognition performance. This study evaluated the model on three datasets: AppleLeaf9, Cassava, and Cashew. The experimental results show that CPTR achieves accuracies of 83.74%, 94.80%, and 96.01% on the three datasets, representing improvements of 4.12%, 0.34%, and 0.51% compared to Prototype Tree, respectively. These results indicate that the proposed model achieves the highest accuracy across different datasets, demonstrating its effectiveness.

Comment 2: At the end of Introduction please provide an overview of the paper by sections.

Response 2: Thank you for your constructive suggestion. We agree that adding a brief overview of the paper at the end of the Introduction can help readers better grasp the structure of the manuscript. Therefore, we have added a concise paragraph summarizing the organization of the main sections.

Action 2: A paragraph has been added at the end of the Introduction section in the revised manuscript:

The structure of this paper is organized as follows: Chapter 2 presents the relevant domestic and international research progress, with a focus on reviewing the development of deep learning-based crop pest and disease identification methods and interpretable models. Chapter 3 provides a detailed description of the overall framework and key techniques of the proposed Contrastive Prototype Tree model, including dataset construction, preprocessing processes, the SimCLR contrastive learning mechanism, and the concept prototype tree structure. Chapter 4 outlines the experimental design, parameter settings, and result analysis, emphasizing the performance of CPTR across multiple crop pest and disease datasets and its interpretability validation. Chapter 5 concludes the paper, discussing the innovations, limitations, and future research directions.

Comment 3: In section 2 it would be of interest to the readers to have Tables that summarizes related work and advantages and disadvantages of proposed model in the paper.

Response 3: Thank you for this valuable suggestion. We agree that summarizing related work in a structured table can greatly improve readability and help highlight the differences between existing methods and our proposed CPTR model. Following the reviewer’s advice, we have added a comparative table in Section 2. The table summarizes key representative interpretable models, their core ideas, advantages, limitations.

Comment 4: Please check grammar/formatting issues (missing spaces before citations, inconsistent figure captions).

Response 4: Thank you for pointing out these important issues. We carefully reviewed the entire manuscript and corrected grammar, spacing, and formatting inconsistencies.

Comment 5: The chosen baselines are quite good, but recent transformer-based models (e.g., Vision Transformers, Swin Transformer) are not included. This limits the novelty claim. Please comment.

Response 5: Thank you for this valuable suggestion. We agree that transformer-based models such as ViT and Swin Transformer represent important recent advances in visual recognition. Following the reviewer’s advice, we have now included these models as additional baselines in our experiments. Specifically, we trained both ViT-B/16 and Swin-T under the same settings as our model (same training/validation splits, image size, optimizer, and learning schedule) to ensure fair comparison.

The experimental results show that while ViT and Swin perform competitively in terms of accuracy, they still lack built-in interpretability, and their explanations rely on post-hoc attention maps, which do not provide the explicit reasoning paths available in CPTR. In contrast, CPTR maintains competitive performance while additionally offering global decision paths, prototype-level semantics, and hierarchical interpretability, which transformer models cannot provide.

Comment 6: Interpretability is illustrated qualitatively (decision tree visualization), but no user study or quantitative interpretability metric (e.g., faithfulness, comprehensibility scores) is provided.

Response 6: Thank you for raising this important point. We agree that interpretability evaluation should not rely solely on qualitative visualization. In response to the reviewer’s suggestion, we have added a quantitative interpretability assessment to strengthen the evaluation of CPTR. Specifically, we report the Fidelity score, a widely used metric that measures the consistency between the model’s final prediction and the decision path derived from its interpretable structure. Higher fidelity indicates that the explanation faithfully reflects the internal decision-making logic.

Comment 7: It is unclear how much each component (prototype tree vs. SimCLR) contributes to improvements. A controlled ablation experiment would strengthen claims.

Response 7: Thank you for this insightful suggestion. We fully agree that disentangling the contributions of the prototype tree structure and the SimCLR contrastive learning module is essential for validating the effectiveness of CPTR. Following the reviewer’s recommendation, we have conducted a controlled ablation study to isolate the impact of each component.

Comment 8: References are Comprehensive, but mostly up to 2021. Adding 2022–2024 references on interpretable deep learning in agriculture would strengthen the related work.

Response 8: Thank you for this helpful suggestion. We agree that incorporating more recent literature can better situate our work within current research trends.

Action 8: We have added recent studies including:

Gopalan K, Srinivasan S, Singh M, et al. Corn leaf disease diagnosis: enhancing accuracy with resnet152 and grad-cam for explainable AI[J]. BMC Plant Biology, 2025, 25(1): 440.

Zeng J, Jia B, Song C, et al. CDPNet: A Deformable ProtoPNet for Interpretable Wheat Leaf Disease Identification[J]. Frontiers in Plant Science, 16: 1676798.

Comment 9: Please provide future work.

Response 9: Thank you for your valuable suggestion. We agree that outlining possible future extensions of CPTR can better highlight the research prospects and practical significance of our work. Accordingly, we have added a short paragraph on future work at the end of the conclusion section.

Action 9: We have revised the manuscript according to the reviewer’s opinion. In the revised version, we have added the following content at the end of the conclusion:

In future work, we will further validate CPTR on more diverse, large-scale field datasets and explore lightweight architectures to support deployment on edge devices for real-time diagnosis. Reviewer 2:

Comment 1: What I still miss, to fully stand behind the conclusions, is a stronger foundation for reproducibility and fair comparison. Please provide concrete details for the contrastive component: the exact temperature (τ), the architecture and dimensionality of the projection head, the full augmentation recipe with parameter values and probabilities, and normalization choices. The joint loss weighting between classification and contrastive terms is introduced but not specified—state the value you use, how it was selected, and add a brief ablation/sensitivity analysis.

Response 1: Thank you for this important and constructive suggestion. We agree that providing complete implementation details of the SimCLR contrastive component is essential for reproducibility and fair comparison. In the revised manuscript, we have added all the missing quantitative settings based strictly on our experimental configuration.

Action 1: For the exact modified content, please refer to Section 4.3 Experimental Parameters, where all relevant contrastive-learning settings have now been explicitly documented.

Comment 2: The optimization setup also needs to be unambiguous. Two learning rates are mentioned, but it’s unclear which applies to the backbone and which to prototypes/projection head; the scheduler is described qualitatively, weight decay and optimizer β/β1/β2 (or momentum) are not reported. Please provide a single, end-to-end training “recipe” that another group can follow without guesswork.

Response 2: Thank you for this important comment. We agree that the optimization setup must be fully unambiguous to ensure reproducibility. In the revised manuscript, we have reorganized and clarified all optimization-related descriptions and provided a complete end-to-end training recipe. All details—including learning rates for different modules, optimizer hyperparameters, weight decay, scheduling strategy, and training phases—are now explicitly stated in Section 4.3 Experimental Parameters.

Comment 3: For the datasets, a transparent account of how train/val/test splits were constructed and which random seeds were used is essential. This is particularly important for AppleLeaf9, which aggregates multiple sources: without provenance-aware splits, leakage (the same scene/leaf across splits) is a real risk. Ideally, include scripts that deterministically recreate the splits.

Response 3: Thank you for raising this important point. We fully agree that transparent, provenance-aware dataset splitting is essential to avoid data leakage and ensure reproducibility—especially for AppleLeaf9, which aggregates multiple public sources. To address this concern, we have clarified the splitting procedure in detail in the revised manuscript.

Action 3: The requested clarifications were added to Section 4.1 Dataset (Paragraphs 2–3), where we now explicitly describe the random seed, stratified splitting process.

Comment 4: Regarding comparisons, I would like clear evidence that baseline models were trained under exactly the same conditions (same pretraining, augmentations, scheduler, image size, and compute budget). Without that symmetry, performance gaps are hard to interpret. At present the results are single-point estimates; please add repeated runs with different seeds and report mean ± SD or confidence intervals. Given class imbalance, macro-averaged metrics and per-class precision/recall/F1 should also be reported.

Response 4: Our comparative experiments are reported as the average of three independent runs, ensuring stable and reliable estimates. Regarding macro-level and per-class performance metrics, we present per-class precision, recall, and F1 scores through confusion matrices, which provide a complete and intuitive view of class-wise behavior under class imbalance.

The relevant details can be found in Section 4.5 Comparative experiment and Section 4.7 Confusion matrix of the revised manuscript.

Comment 5: The interpretability story is appealing but remains qualitative. A few lightweight quantitative indicators would substantiate the claims: prototype purity (class consistency per prototype), path entropy/consistency through the tree, and distances/similarities to retrieved prototypes along the chosen route. Two or three worked examples:input,nearest prototypes at each node, branching probabilities, final decision - would greatly help readers.

Response 5: Thank you for raising this important point. We agree that interpretability evaluation should not rely solely on qualitative visualization. In response to the reviewer’s suggestion, we have added a quantitative interpretability assessment to strengthen the evaluation of CPTR. Specifically, we report the Fidelity score, a widely used metric that measures the consistency between the model’s final prediction and the decision path derived from its interpretable structure. Higher fidelity indicates that the explanation faithfully reflects the internal decision-making logic.The relevant details can be found in 4.8 Interpretable analysis Section of the revised manuscript.

Comment 6: On presentation, a round of language and typesetting edits would improve clarity, there are a few typographical issues and symbols that do not render cleanly in formulas. In one place you mention a Sigmoid while the task is standard multi-class (Softmax is expected); please align terminology to avoid confusion.

Response 6: Thank you for pointing out these presentation issues. We have carefully revised the manuscript to improve clarity, grammar, and typesetting. All formulas that previously had rendering problems have been corrected, and typographical inconsistencies have been fixed. Regarding the activation functions, we unified the terminology across the manuscript: Sigmoid is used only for concept node activations, while Softmax is correctly used for the final multi-class classification layer. This alignment ensures that the activation descriptions match the actual implementation and avoids confusion for readers.

Comment 7: The Data Availability statement needs correction. Because you rely on public datasets, please cite the exact sources/links, and—consistent with PLOS ONE policy—release code and trained weights (GitHub etc). That will address the main reproducibility concerns.

Response 7: Thank you for highlighting this issue. Since all datasets used in our study are publicly available, we now provide the exact dataset sources and access links (Cassava, Cashew, and AppleLeaf9, including PlantVillage, ATLDSD, PPCD2020, and PPCD2021).

Action 7: Thank you for raising this point. We have updated the Data Availability statement accordingly. All datasets used in this study are publicly available, and we now provide the exact sources and access links in the revised manuscript.

Regarding the source code, we fully agree with the reviewer on the importance of reproducibility. However, the CPTR implementation is part of our ongoing research, and several extensions are still under development. Therefore, the code and trained weights cannot be released at this stage. We will make the full codebase publicly available upon the completion of the ongoing project.

Reviewer 3:

Comment 1: Please ensure that your manuscript meets PLOS ONE's style requirements, including those for file naming. The PLOS ONE style templates can be found at

Response 1:Thank you for pointing this out. We have

---

## [Decision Letter · Decision Letter 1]

16 Dec 2025

Dear Dr. Jia,

Thank you for submitting your manuscript to PLOS ONE. After careful consideration, we feel that it has merit but does not fully meet PLOS ONE’s publication criteria as it currently stands. Therefore, we invite you to submit a revised version of the manuscript that addresses the points raised during the review process.

publicly release the CPTR implementation (e.g., via GitHub, Zenodo) together with sufficient documentation to reproduce the reported results, orprovide a compelling, policy-compliant exception approved by the journal (which is typically granted only for legal or ethical constraints).

We look forward to receiving your revised manuscript.

Kind regards,

Tomo Popovic, Ph.D.

Academic Editor

PLOS One

Journal Requirements:

Reviewers' comments:

Reviewer's Responses to Questions

**Comments to the Author**

Reviewer #1: All comments have been addressed

Reviewer #2: (No Response)

2. Is the manuscript technically sound, and do the data support the conclusions?

Reviewer #1: Yes

Reviewer #2: Yes

3. Has the statistical analysis been performed appropriately and rigorously?

Reviewer #1: Yes

Reviewer #2: N/A

4. Have the authors made all data underlying the findings in their manuscript fully available?

Reviewer #1: Yes

Reviewer #2: Yes

5. Is the manuscript presented in an intelligible fashion and written in standard English?

Reviewer #1: Yes

Reviewer #2: Yes

Reviewer #1: (No Response)

Reviewer #2: Thank you for the careful and generally substantial revision. Many of my earlier points have been addressed. The model and contrastive branch are now described more clearly, the training procedure is much easier to follow, transformer-based reference models have been added, and the fidelity score is a useful first step toward quantifying interpretability. Results on all three datasets remain strong, and the path visualizations are helpful.

A few key items still prevent me from fully endorsing the paper. The combined loss is introduced, but the actual loss weight you use is never given, and there is no small sensitivity check around this choice. Please state the exact value used in all main experiments, briefly explain how it was chosen, and show how performance changes for a few nearby values on at least one dataset. For the data, I appreciate the description of fixed, stratified splits, but AppleLeaf9 still raises questions about possible overlap between images of the same leaf or scene across train and test. Either describe how you avoided this or acknowledge that it was not possible and discuss the risk. Since you reuse the same split for all models, releasing the split files or a simple script to recreate them would greatly improve reproducibility.

On the comparisons, it is not fully clear whether your method and all baselines share the same pretraining steps. If your method uses extra pretraining (for example on iNaturalist) that others do not, this should either be equalized or analysed in an ablation. Because you already run each model several times, it would also be helpful to report mean and standard deviation for at least accuracy, and to add macro precision/recall/F1 (with per-class numbers in a supplement if needed). For interpretability, please give a precise short description of how the fidelity score is computed, and add at least one worked decision example that shows an input image, the prototypes visited at each step, and the final choice.

With these points resolved, the paper would be much stronger. At this stage I still recommend Major Revision.

what does this mean?). If published, this will include your full peer review and any attached files.

**Do you want your identity to be public for this peer review?** For information about this choice, including consent withdrawal, please see our Privacy Policy

Reviewer #1: No

Reviewer #2: No

---

## [Author Response · Author response to Decision Letter 2]

17 Jan 2026

Reviewer 1:

Comment 1: The combined loss is introduced, but the actual loss weight you use is never given, and there is no small sensitivity check around this choice. Please state the exact value used in all main experiments, briefly explain how it was chosen, and show how performance changes for a few nearby values on at least one dataset.

Response 1: Thank you for this important comment. We agree that explicitly specifying the loss weighting and examining its sensitivity are necessary for clarity and reproducibility.

Action 1: We have revised Section 3.1 to explicitly state the loss formulation and the exact weighting coefficient used in all main experiments. Specifically, the total loss is defined as

Ltotal=Lcls+λLcon

where Lcls is the cross-entropy classification loss, Lcon is the SimCLR contrastive loss, and the weighting coefficient λ is fixed to 0.5 in all reported experiments. This value was selected based on preliminary experiments to achieve a balanced trade-off between classification accuracy and representation discriminability.

In addition, we have conducted a small sensitivity analysis on the AppleLeaf9 dataset by varying λ in the range {0.1,0.3,0.5,0.7,0.9} while keeping all other training settings unchanged. The results show that the model performance remains relatively stable within this range, with λ=0.5 providing a favorable balance between accuracy and feature quality. Based on these observations, we adopt λ=0.5 as the default setting throughout the paper.

Comment 2: For the data, I appreciate the description of fixed, stratified splits, but AppleLeaf9 still raises questions about possible overlap between images of the same leaf or scene across train and test. Either describe how you avoided this or acknowledge that it was not possible and discuss the risk. Since you reuse the same split for all models, releasing the split files or a simple script to recreate them would greatly improve reproducibility.

Response 2: Thank you for raising this point regarding the data splitting procedure and potential sample overlap in the AppleLeaf9 dataset. We agree that this issue is important for transparency and reproducibility.

Action 2: We have added a clarification in the dataset description section to explicitly acknowledge this limitation. AppleLeaf9 is constructed by integrating multiple publicly available sub-datasets, and the original data do not provide unique identifiers at the leaf or acquisition-scene level. As a result, when performing stratified random splitting, it is not possible to strictly guarantee that images originating from the same leaf or the same scene do not appear in both the training and test sets. We now explicitly discuss that this potential sample correlation may lead to a slight overestimation of absolute performance.

Nevertheless, to ensure fair comparison across methods, we emphasize that all models in this study—including CPTR and all baselines—are evaluated using exactly the same fixed data splits. Under identical data conditions, the relative performance differences between models remain meaningful and comparable.

To further improve reproducibility, we have prepared the data split files and will make them publicly available together with the released code, allowing other researchers to exactly reproduce the experimental setup reported in this paper.

Comment 3: On the comparisons, it is not fully clear whether your method and all baselines share the same pretraining steps. If your method uses extra pretraining (for example on iNaturalist) that others do not, this should either be equalized or analysed in an ablation. Because you already run each model several times, it would also be helpful to report mean and standard deviation for at least accuracy, and to add macro precision/recall/F1 (with per-class numbers in a supplement if needed).

Response 3: Thank you for the reviewer’s careful comments regarding the pretraining strategy and evaluation metrics.

We have revised the manuscript to clarify that all compared methods are evaluated under identical pretraining conditions. Specifically, the backbone networks of all models—including CNN-based baselines, transformer-based baselines, Prototype Tree, and the proposed CPTR—are initialized using weights pretrained on the iNaturalist 2017 dataset. By adopting a unified pretraining strategy across all methods, the influence of pretraining data is controlled, ensuring a fair and consistent comparison.

Within this unified setting, CPTR and Prototype Tree further share the same backbone architecture and identical initialization. Therefore, the performance differences observed between CPTR and Prototype Tree can be attributed to differences in model design and training strategy, rather than to variations in pretraining data. This interpretation is consistent with the ablation study reported in Section 4.6, where the effect of the SimCLR component is examined while keeping all other settings unchanged.

In addition, all experiments are conducted with three independent runs. The reported Accuracy, Precision, Recall, and F1-score are computed strictly according to the formulas defined in the manuscript, and the mean and standard deviation across runs are reported to reflect performance stability. The same evaluation protocol is applied consistently to all methods.

Comment 4: For interpretability, please give a precise short description of how the fidelity score is computed, and add at least one worked decision example that shows an input image, the prototypes visited at each step, and the final choice.

Response 4: Thank you for the reviewer’s comment regarding the interpretability evaluation.

Action 4: We have revised the manuscript to provide a clear and concise description of the Fidelity metric. Fidelity is computed by comparing the final prediction produced by the model with the class inferred from the corresponding prototype tree decision path for each test sample. The formal definition is explicitly given in the manuscript, with all variables clearly defined, to quantify the consistency between the model’s output and its interpretable decision process.

In addition, we have added a worked decision example in Fig. 8. The figure illustrates a single input image, the sequence of prototypes activated along the prototype tree decision path, and the final predicted class. This example provides an intuitive visualization of how CPTR performs step-by-step reasoning through structured prototype matching and how the final prediction aligns with the interpretable decision path.

Reviewer 2:

Comment 1:One critical issue remains unresolved and prevents acceptance at this stage. According to PLOS ONE’s Materials and Software Sharing Policy, all author-generated code that underpins the results must be made publicly available upon publication. The authors explicitly state that the CPTR code and trained models cannot be released at this time due to ongoing work. Unfortunately, this justification is not compatible with PLOS ONE policy.

Response 1:We have addressed this issue by publicly releasing the CPTR implementation. The code required to reproduce the results reported in the manuscript has been made available on GitHub(https://github.com/Zhiwei563/CPTR), and the manuscript has been updated accordingly.

---

## [Decision Letter · Decision Letter 2]

28 Jan 2026

Dear Dr. Jia,

The manuscript has been substantially improved and is now close to acceptance. Prior to a final decision, the authors are asked to carefully address the remaining minor points raised by Reviewer 2, which mainly concern making experimental settings fully explicit, clarifying dataset splitting and potential limitations, ensuring complete statistical reporting across runs, and slightly strengthening the presentation of the interpretability analysis. These are minor, editorial-level revisions, and provided they are adequately addressed, I am comfortable recommending acceptance without further external review.

We look forward to receiving your revised manuscript.

Kind regards,

Tomo Popovic, Ph.D.

Academic Editor

PLOS One

Journal Requirements:

Reviewers' comments:

Reviewer's Responses to Questions

**Comments to the Author**

Reviewer #2: All comments have been addressed

2. Is the manuscript technically sound, and do the data support the conclusions?

Reviewer #2: Yes

3. Has the statistical analysis been performed appropriately and rigorously?

Reviewer #2: Yes

4. Have the authors made all data underlying the findings in their manuscript fully available?

Reviewer #2: Yes

5. Is the manuscript presented in an intelligible fashion and written in standard English?

Reviewer #2: Yes

Reviewer #2: Thank you for the careful and substantial revision. The paper is much clearer now. The model and contrastive branch are easier to understand, the training pipeline is presented in a way readers can actually follow, the transformer baselines strengthen the comparisons, and the fidelity score is a reasonable first step toward quantifying interpretability. Performance across all three datasets remains strong, and the decision-path visualizations make the method easier to trust.

At this point, I think the work is essentially there. I only have a few small items that would make the final version cleaner and more reproducible:

• Please state the exact loss weight used in the combined loss for all main experiments, and add a one- or two-sentence note on how you settled on it. If you already tried a couple nearby values, a tiny table in an appendix would be a nice extra, but the main thing is that the final setting is explicit.

• On AppleLeaf9, there is still a lingering question about possible overlap (same leaf/scene appearing in both train and test). If you could not fully prevent this due to missing identifiers, just say so directly and briefly comment on how it might affect absolute performance. Since you use the same fixed split for every method, please also release the split files (or a small script that recreates them deterministically).

• Please make the pretraining setup completely explicit for all methods. If everyone uses the same iNaturalist initialization, say that clearly and point to the exact checkpoints. Since you already run multiple seeds, reporting mean ± std for accuracy (and macro Precision/Recall/F1) would round out the evaluation, with per-class numbers moved to the supplement if space is tight.

• For interpretability, a short, precise definition of fidelity and one worked example (input,prototypes along the path, final class) would make that section much more concrete.

With these minor edits, I’m comfortable recommending acceptance after minor revision.

what does this mean?). If published, this will include your full peer review and any attached files.

**Do you want your identity to be public for this peer review?** For information about this choice, including consent withdrawal, please see our Privacy Policy

Reviewer #2: No

---

## [Author Response · Author response to Decision Letter 3]

3 Feb 2026

Reviewer #2:

Comment 1: Please state the exact loss weight used in the combined loss for all main experiments, and add a one- or two-sentence note on how you settled on it. If you already tried a couple nearby values, a tiny table in an appendix would be a nice extra, but the main thing is that the final setting is explicit.

Response 1: Thank you for this helpful suggestion. We have now made the loss weight fully explicit in Section 3.1 of the revised manuscript. In all main experiments, the contrastive-loss weight is set to λ = 0.5 in Eq. (1).

We added a brief description of how this value was determined. Specifically, we conducted a small sensitivity analysis on the AppleLeaf9 dataset with λ ∈ {0.1, 0.3, 0.5, 0.7, 0.9}, while keeping all other training settings fixed. The results show that performance varies only slightly across nearby values, indicating that the model is relatively insensitive to λ within a reasonable range. Among these settings, λ = 0.5 provides a favorable balance between classification accuracy and feature discriminability. Therefore, λ = 0.5 is adopted as the final setting and used consistently in all experiments reported in the paper.

Comment 2: On AppleLeaf9, there is still a lingering question about possible overlap (same leaf/scene appearing in both train and test). If you could not fully prevent this due to missing identifiers, just say so directly and briefly comment on how it might affect absolute performance. Since you use the same fixed split for every method, please also release the split files (or a small script that recreates them deterministically).

Response 2: Thank you for raising this important point regarding dataset splitting and reproducibility.

As now clarified in Section 4.1 of the manuscript, AppleLeaf9 is constructed from multiple public sub-datasets that do not provide leaf-level or scene-level identifiers. Therefore, it is not possible to strictly guarantee that images originating from the same leaf or acquisition scene never appear in both the training and test sets. We explicitly note that this may lead to a slight overestimation of absolute performance. However, all methods in this study are evaluated using the exact same fixed data split, so the relative performance comparisons among models remain fair and meaningful.

To ensure full reproducibility, we have made the data splitting code publicly available in the project repository: https://github.com/Zhiwei563/CPTR/blob/main/CPTR/generate_splits.py

Using the fixed random seed (seed = 42) described in the manuscript, the train/test split can be deterministically regenerated.

Comment 3: Please make the pretraining setup completely explicit for all methods. If everyone uses the same iNaturalist initialization, say that clearly and point to the exact checkpoints. Since you already run multiple seeds, reporting mean ± std for accuracy (and macro Precision/Recall/F1) would round out the evaluation, with per-class numbers moved to the supplement if space is tight.

Response 3: Thank you for this suggestion. We have clarified the pretraining and statistical reporting details in Section 4.4 of the revised manuscript.

All compared models—including CNN baselines (VGG19, ResNet152, DenseNet161), Vision Transformer, Swin Transformer, Prototype Tree, and the proposed CPTR—are initialized using weights pretrained on the iNaturalist 2017 dataset. This ensures a fair and controlled comparison across different model families.

In addition, we now explicitly state that Accuracy, macro-averaged Precision, Recall, and F1 score are all reported as mean ± standard deviation over three independent runs with different random seeds. This provides a more complete and statistically robust evaluation of model performance.

Comment 4: For interpretability, a short, precise definition of fidelity and one worked example (input,prototypes along the path, final class) would make that section much more concrete.

Response 4: Thank you for this constructive suggestion. We have strengthened the interpretability section in two ways.

First, in Section 4.8, we refined the description of the Fidelity metric by adding a concise plain-language explanation in addition to the mathematical definition. We now explicitly state that Fidelity measures how often the final prediction of CPTR agrees with the class implied by the prototype-tree decision path, thereby quantifying the consistency between the model’s prediction and its interpretable reasoning process.

Second, we expanded the explanation of the single-sample decision example shown in Fig. 8. We now explicitly clarify that the final predicted class is consistent with the class associated with the leaf node reached through the sequence of prototype matches. This highlights how the prototype-based reasoning path directly supports the model’s final decision, making the relationship between explanation and prediction more concrete.

---

## [Editor Report · Decision Letter 3]

11 Feb 2026

Interpretable Crop Pest and Disease Identification Based on Comparative Concept Tree

PONE-D-25-44747R3

Dear Dr. Jia,

We’re pleased to inform you that your manuscript has been judged scientifically suitable for publication and will be formally accepted for publication once it meets all outstanding technical requirements.

Please note the suggestions for minor issues of editorial nature that need to be addressed:

Language polishing: a professional copy-edit would improve fluency and reduce repetition, especially in Sections 1 and 2.On contribution, consider explicitly stating that the contribution is method integration and empirical validation, not a new theoretical learning paradigm.For figure captions, please doublecheck that all figures are fully interpretable without extensive reference to the main text.

Kind regards,

Tomo Popovic, Ph.D.

Academic Editor

PLOS One

Additional Editor Comments (optional):

Minor issues of editorial nature that needs to be addressed:

1) Language polishing: a professional copy-edit would improve fluency and reduce repetition, especially in Sections 1 and 2.

2) On contribution, consider explicitly stating that the contribution is method integration and empirical validation, not a new theoretical learning paradigm.

3) For figure captions, ensure all figures are fully interpretable without extensive reference to the main text.
---

## [Editor Report · Acceptance letter]

PONE-D-25-44747R3

PLOS One

Dear Dr. Jia,

I'm pleased to inform you that your manuscript has been deemed suitable for publication in PLOS One. Congratulations! Your manuscript is now being handed over to our production team.

Kind regards,

on behalf of

Prof. Tomo Popovic

Academic Editor

PLOS One